# Tropical precipitation response to anthropogenic climate change in recent decades

Ligin Joseph [1,4] ✉, Pascal Terray [2,4], K. P. Sooraj[3] & Sébastien Masson[2]

Tropical rainfall plays a central role in the climate system, shaping ecosystems and societies. Here we show that recent tropical rainfall changes are primarily driven by spatial shifts in atmospheric circulation rather than thermodynamic processes, and cannot be explained by the "Wet Get Wetter" or "Warm Get Wetter" paradigms. Observations reveal a northward shift in precipitation with wetting in the western and northern equatorial Pacific, northern Indian region, and drying south of the equator in the Pacific and South America. These trends coincide with a La Niña-like sea surface temperature pattern, strengthened Walker circulation, Southern Ocean cooling, enhanced land-sea and inter-hemispheric thermal gradients, and intensification of the Indo-Pacific warm pool. Climate models largely miss the first three features, projecting instead a reduced equatorial Pacific sea surface temperature gradient, but capture large-scale thermal gradients and Indo-Pacific warm pool changes. We show that amplified land-sea thermal contrast and Indo-Pacific warm pool intensification reproduce the observed circulation and rainfall changes. Coupled sensitivity experiments further confirm that land warming and ongoing desertification in the Northern Hemisphere act as active drivers of current tropical hydroclimate changes, challenging ocean-centric assumptions in current climate models.

The world is warming at an unprecedented pace. According to the State of the Climate 2024 report from the World Meteorological Organization, the global mean near-surface air temperature in 2024 was 1.55 °C above the pre-industrial average, making it the hottest year on record[1]. Rising temperatures have far-reaching impacts, including land and marine heat waves, glacier melt, sea-level rise, more frequent extreme weather events, and ecosystem disruptions[2–7]. Among these, changes in precipitation stand out because of their direct and profound effects on human societies, particularly in the tropics. Yet the relationship between climate change and rainfall is far more complex than that with temperature[8,9]. Warming can intensify flooding in some regions while driving droughts in others, and in many areas, trends in mean precipitation remain obscured by natural variability[6]. Understanding how climate change alters precipitation is therefore critical

but remains scientifically challenging since current coupled climate models still suffer from large Sea Surface Temperature (SST) and precipitation biases in the tropics[10–13] which may mask or distort the underlying climate-change signal[14].

Different theories have been proposed to explain how precipitation responds to climate change, including the "Wet Get Wetter" (WeGW), the "Direct effect of $CO_2$ forcing" (DeCO$_2$), and the "Warm Get Wetter" (WaGW) paradigms[15–19]. These paradigms, or their hybrids, provide skillful explanations for future regional tropical precipitation changes over oceans under centennial-scale climate change and in idealized experiments forced with abrupt or gradual $CO_2$ increase. However, recent studies demonstrate that these frameworks are not directly applicable over land in their original forms[20–24], and further highlight the importance of land-sea thermal contrasts, which alter

[1]School of Ocean and Earth Science, University of Southampton, Southampton, United Kingdom. [2]Sorbonne Universités (UPMC, Univ Paris 06)-CNRS-IRD-MNHN, LOCEAN Laboratory, 4 place Jussieu, Paris, France. [3]Centre for Climate Change Research, Indian Institute of Tropical Meteorology, Ministry of Earth Sciences, Pune, India. [4]These authors contributed equally: Ligin Joseph, Pascal Terray. ✉e-mail: l.joseph@soton.ac.uk

atmospheric circulation and drive regional rainfall variations independently of $CO_2$ radiative effects or SST patterns[20,24–26].

Taken together, these theories and their recent refinements that incorporate the influence of land-sea thermal contrasts offer valuable frameworks for understanding future projections of tropical precipitation and circulation changes in climate models. However, it remains unclear whether they can adequately explain the current observed tropical precipitation and atmospheric circulation trends, which is the main focus of this paper.

The tropical Pacific strongly influences global precipitation patterns across a range of time scales. Therefore, it is crucial to consider forced changes in the equatorial Pacific when assessing tropical precipitation and atmospheric circulation responses to climate change. In this context, the response of the equatorial Pacific's zonal SST gradient and the associated Walker circulation to global warming remains a subject of active debate[27–34]. The WeGW paradigm suggests a weakening of atmospheric circulations, including the Walker and Hadley cells, due to increased atmospheric moisture and reduced radiative cooling. However, observational data over recent decades indicate a strengthening of the Walker circulation, despite significant surface warming[29,32,33]. Previous studies using attribution and idealized experiments have further shown that this apparent contradiction is closely linked to the behavior of the zonal SST gradient in the equatorial Pacific and the Indian Ocean, and to the transient nature of the Walker circulation strengthening[24,31,35]. Although many Coupled Model Intercomparison Project (CMIP) coupled models simulate a decrease in this equatorial zonal SST gradient during recent decades under anthropogenic forcing, observations have shown an increase during recent decades, leading to a more La Niña-like state[29,31,33,34].

Several studies have provided comprehensive reviews of this discrepancy[33–36]. They explore hypotheses involving atmospheric stability, evaporative damping, the ocean thermostat, the "iris effect", and internal variability. These mechanisms can predict both the weakening and the strengthening of the Pacific zonal SST gradient under global warming. However, other studies suggest that these discrepancies are unlikely to be explained solely by internal variability or transient features[28–30]. Model biases in accurately representing cloud feedback or El Niño-Southern Oscillation (ENSO) mechanisms may also contribute to this discrepancy. In addition, the Southern Ocean has cooled in recent decades, a feature not well reproduced by most climate models, and this cooling may influence the equatorial Pacific SST gradient[33,37].

Consistent with the WeGW paradigm, Shrestha and Soden[38] argues that the global mean atmospheric overturning circulation has already weakened in recent decades in both simulations and observations. In CMIP historical simulations, this weakening is primarily due to a weaker Pacific Walker circulation. However, in Atmospheric Model Intercomparison Project (AMIP) simulations, where the Pacific Walker circulation shows a strengthened signal, the global mean overturning circulation still weakens. This suggests that a weakening of the global circulation does not require a weakening of the Walker circulation alone[24,39]. Furthermore, using observational records of precipitation and specific humidity, Shrestha and Soden[38] demonstrated that the global-mean overturning circulation is also weakening in observations. The magnitude of this observed weakening is comparable to that seen in the AMIP multi-model mean, though smaller than in the CMIP models.

Based on these studies, it is evident that the precipitation response to global warming is a complex phenomenon, and there remains significant debate over the strength and nature of atmospheric circulation changes, especially during recent decades. Most previous studies have focused primarily on future climate projections and idealized experiments. In contrast, how the water cycle has actually changed over recent decades remains relatively underexplored. With the availability of long-term, satellite-era records and high-quality observationally constrained reanalysis datasets, we are now in a better position than ever to examine how precipitation has responded to ongoing climate change. This raises many critical questions: Can observed precipitation trends over recent decades be explained by existing theoretical frameworks such as WeGW, DeCO2, and WaGW? Or do these trends point toward emerging[34] or alternative mechanisms, for example, those associated with enhanced land-sea thermal contrasts[24,25,40]? To address these questions, this study analyzes observational and reanalysis data from 1979 to 2024, CMIP6 historical simulations, and several dedicated coupled sensitivity experiments, with a focus on understanding the current response of tropical precipitation to global warming.

Here, we show that tropical precipitation changes during the satellite era are primarily driven by large-scale atmospheric circulation spatial shifts in the Indo-Pacific region rather than thermodynamic processes. Furthermore, our results indicate that land warming and ongoing desertification in the Northern Hemisphere act as active drivers of observed tropical hydroclimate changes, challenging ocean-centric assumptions in current climate models.

## Results

We organize the Results section as follows. First, we present spatial trends in key variables like precipitation, 2-meter air temperature (t2m), SST, 850-hPa winds, and upper-tropospheric (200-hPa) velocity potential, over 1979-2024 in reanalyses, observations, and CMIP6 historical simulations. Second, we decompose precipitation changes into dynamic and thermodynamic components to assess the relative role of circulation versus moisture constraints on the precipitation change, and we show that the recent observed rainfall trends are mostly driven by dynamic processes. Third, we analyze eight surface temperature indices in observations and simulations, partition their trends and residuals, and use spatial regressions to demonstrate that enhanced land-sea contrast and Indo-Pacific warm pool intensification drive tropical atmospheric circulation and rainfall trends over 1979–2024 period in observations. Finally, we compare observations with targeted coupled-model sensitivity experiments that artificially amplify the land-sea thermal gradient, evaluating its ability to reproduce the observed precipitation, SST, wind, and upper-level velocity potential trend patterns.

### Global climate trends

The trend maps of precipitation, t2m, SST, 850-hPa winds, and 200-hPa velocity potential from ERA5 are shown in Fig. 1. ERA5 is used as the primary observational reference in this study because it reliably reproduces the key characteristics of recent precipitation changes[41]. However, to assess whether these trends are robust or specific to ERA5, we first compare them with trends derived from observational-based (GPCP[42] for precipitation, Berkeley Earth[43] for t2m, OISST[44] for SST) products and other reanalysis products (NCEP2[45] for wind and velocity potential), shown in Supplementary Fig. 1. ERA5 broadly agrees with other observational and reanalysis products, especially for Pacific precipitation, polar amplification, land-sea thermal contrast, La-Niña-like SST trends, Southern Ocean cooling, and Pacific Walker circulation strengthening. However, differences remain, notably stronger drying/wetting in ERA5 vs. GPCP, discrepancies over East Asia/Africa, and contrasting SST trends in the equatorial Pacific. Hereafter, we will discuss in more detail only the robust trends across the datasets and assess whether CMIP6 simulations capture them, particularly precipitation and temperature trends.

The robust precipitation trends reveal both significant wetting and drying patterns across the globe (Fig. 1a). The strongest trends are found over the tropical Pacific Ocean on either side of the equator, with increasing precipitation along the western equatorial Pacific, the Maritime Continent, and the Intertropical Convergence Zone (ITCZ). A notable drying trend is observed just south of the Northern ITCZ, and

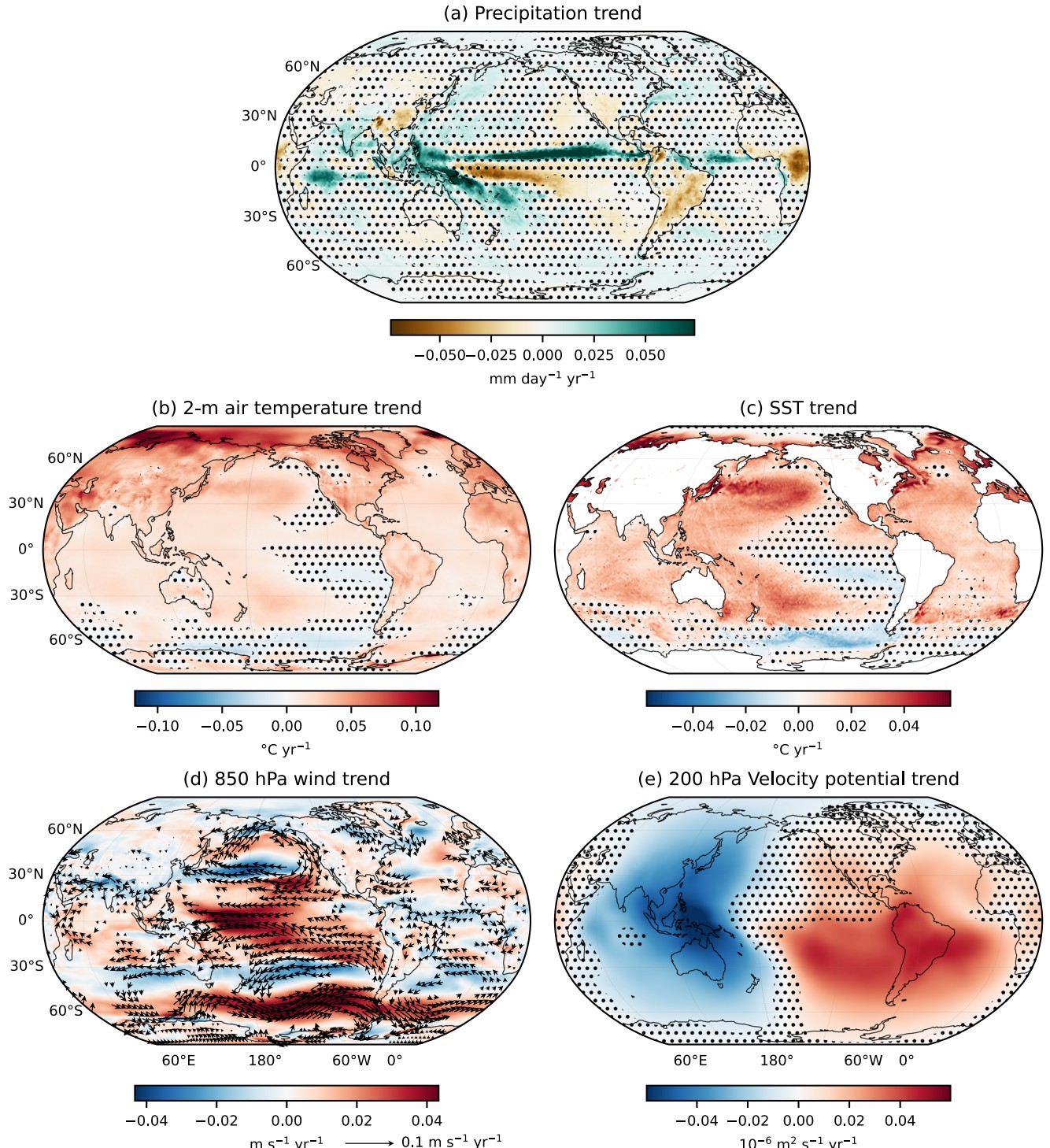

**Fig. 1 | Global climate trends (1979-2024).** Trends in (**a**) precipitation (mm day$^{-1}$ yr$^{-1}$), **b** 2-meter air temperature (t2m; °C yr$^{-1}$), **c** Sea Surface Temperature (SST; °C yr$^{-1}$), **d** 850-hPa wind vector and speed (m s$^{-1}$ yr$^{-1}$), and **e** 200-hPa velocity potential (m$^2$ s$^{-1}$ yr$^{-1}$) using ERA5. Dots mark regions with statistically insignificant trends (at 90% confidence level). In (**d**), arrows show wind vector trends, plotted only if at least one component is significant at 90% confidence level.

the South Pacific Convergence Zone (SPCZ) is more tilted in the southeast direction. Overall, these changes suggest a strengthening and narrowing of the ITCZ in the Pacific and Atlantic Oceans (commonly referred to as the "deep-tropics squeeze"), which is also found in independent atmospheric infrared sounder observations[46] and likely driven by global warming[47]. An increasing trend in precipitation is also evident over the Indian regions in both ERA5 and GPCP. Meanwhile, regions such as South America also exhibit a robust drying trend.

We next compared these observed precipitation trends with both CMIP6 Multi-Model Mean (MME) historical rainfall trends over 1979-2024 and the late-century CMIP6 projections, as shown in Supplementary Fig. 2. This allows us to assess whether present-day changes anticipate the far-future response seen in CMIP models. First, Supplementary Fig. 2b shows that the CMIP6 models' projections and simulations retain a pronounced double-ITCZ bias, which is a long-standing known bias in all past CMIP simulations[12,48]. Next, it is clear

that the rainfall trend pattern over 1979-2024 in CMIP6 historical simulations is almost similar to the century-scale response in the CMIP6 projections (spatial correlation of 0.7, Supplementary Fig. 2c, d). Both bear a strong similarity to an El Niño-like rainfall pattern (see Supplementary Fig. 3a), which is partly muted by the double-ITCZ bias. This El Niño-like rainfall pattern in CMIP6 simulations is markedly different from the observed rainfall trend pattern (Fig. 1a). As an illustration, the large-scale drying south of the equator in the Pacific and the large increase in precipitation over the Maritime Continent and along the North ITCZ are not reproduced at all by the CMIP6 MME of historical simulations and projected changes. Importantly, the observed rainfall trend pattern in CMIP6 is also different from the La Niña-like rainfall pattern (Supplementary Fig. 3b).

Taken together, these large discrepancies indicate that present-day rainfall patterns in observations are not a simple early manifestation of the late-century response shaped by the WeGW, DeCO$_2$ and WaGW mechanisms. Instead, they likely reflect a transient adjustment, a manifestation of internal variability or, more drastically, a systematic inability of current climate models to reproduce the current forced signals[28–30]. This highlights the urgent need to diagnose in more detail the origins of observed rainfall trends.

We will now focus on other climate variables, which are physically related to precipitation. The t2m trends clearly reflect signatures of global warming, with the majority of the globe showing warming trends with only some localized, yet robust, regional cooling, especially over the southeastern Pacific Ocean, off the equator, and the Southern Ocean (Fig. 1b). The maximum warming trend is seen over northern latitudes-over the Arctic region in ERA5, but over northern Eurasia and Greenland in observation (Supplementary Fig. 1b) which is a robust signal of climate change and is usually referred to as Arctic or Polar Amplification[49,50]. Furthermore, it is observed that the warming over land is systematically and substantially greater than over the ocean, reflecting the well-documented land-sea warming contrast in response to increased atmospheric CO$_2$ concentrations[51–53]. As this contrast is found in both observations, and transient and near-equilibrium simulations[54,55], it cannot be attributed solely to transient differences in thermal inertia and ocean heat uptake. Instead, it may arise from effective radiative forcing, atmospheric energy transport anomalies, and distinct feedbacks involving enhanced evaporative cooling over the ocean, differences in lapse-rate changes between land and sea, and cloud-radiation interactions that modify surface short-wave fluxes[52–55]. The t2m trend further shows a pronounced inter-hemispheric asymmetry, with stronger warming in the Northern Hemisphere (NH), consistent with both its larger land fraction and the fact that the above feedbacks contributing to the land-sea contrast are more effective in the NH[51,56].

The SST trends also show a global warming signature (Fig. 1c). In the tropics, this warming signal is dominated by the extension and intensification of the Indo-Pacific warm pool, which has almost doubled in size since 1900[57] and draws warm-moist air from the tropical Pacific Ocean into the Maritime Continent on an annual average (Fig. 1d). The SST trends in the tropical Pacific resemble a La Niña-like pattern, with warming in the west and cooling in the east. Yet, the eastern equatorial cooling trends are statistically insignificant and non-robust (Fig. 1b, c, and Supplementary Fig. 1b, c), even though the resulting modulation of the Pacific equatorial gradient has attracted a lot of attention in the literature[27,29,31–33,36]. Significant cooling in the south subtropical and southern Pacific, connected to the tropical eastern Pacific cooling, is also found. The Southern Ocean cooling and its possible drivers have also been widely studied, with explanations ranging from natural climate variability[58], stratospheric ozone depletion[59], to Antarctic meltwater influx[60].

This peculiar observed pattern in the Pacific has attracted growing attention, as CMIP climate models generally project an El Niño-like SST trend under future warming scenarios[36]. Moreover, many models

struggle to reproduce the Southern Ocean cooling in their historical simulations (Supplementary Fig. 2e, f). Kang et al.[37] have investigated this discrepancy and have suggested potential links between Southern Ocean cooling and the observed Pacific SST trends. Importantly, these varying patterns in t2m and SST trends may influence regional precipitation distributions differently in observations and simulations, a relationship explored further in subsequent sections.

Consistent with the observed SST trends in the Pacific, the 850-hPa wind trends suggest a strengthening of the surface branch of the Walker circulation in the tropical Pacific (Fig. 1d, and Supplementary Fig. 1d). To validate this claim, we estimated the strength of the Walker circulation in ERA5 by an index based on zonal sea-level pressure variations in the tropical Pacific. More precisely, this Walker Circulation Index (WCI) is defined by the zonal contrast of sea-level pressure between the eastern (5°S-5°N, 160°-80°W) and western (5°S-5°N, 80°-160°E) equatorial Pacific and is commonly used to monitor the trends of the Walker circulation in both observations and CMIP simulations[33,61]. The WCI shows a positive linear trend of 1.397 Pa per year in ERA5 (significant at the 90% confidence level), which contradicts theoretical expectations and future projections but agrees with previous studies that also report this strengthening, as discussed in the Introduction. This enhanced Walker circulation is further supported by significant trends in the 200-hPa velocity potential fields, which show both a large strengthening of upper-level divergence over the Indo-Pacific warm pool and of upper-level convergence, particularly in the southeast Pacific (Fig. 1e). Importantly, like the precipitation trend pattern, this 200-hPa velocity potential pattern differs markedly from the ones observed during El Niño and La Niña events (Supplementary Fig. 3c, d) as the upper-level divergence pole is shifted westward. The upper-level convergence is fully asymmetric with respect to the equator in the observed 200-hPa velocity potential trends (Fig. 1e). In comparison, the ENSO upper-level velocity potential patterns are fully symmetric with respect to the equator (Supplementary Fig. 3c, d).

A similar trend analysis for boreal summer (June-September) and boreal winter (December-March) exhibits comparable large-scale features, with some important distinctions (Supplementary Fig. 4). The Indo-Pacific warm pool intensifies in both seasons, but the lower- and upper-level atmospheric trend patterns are markedly different between the two seasons. During boreal summer, an asymmetric teleconnection pattern develops between the Indian region and the southeast Pacific with a continuous band of southeasterly 850-hPa wind anomalies stretching from the southeast Pacific to the north-eastern African coast and a tilted dipole in the 200-hPa velocity potential trends with upper-level divergence over Africa and the Indian domain, and opposing upper-level convergence over the Southeast Pacific. By contrast, during boreal winter, the symmetric zonal circulation is enhanced with zonal wind convergence towards the Indo-Pacific warm pool at the surface (Supplementary Fig. 4h) and upper-level divergence over the Maritime Continent, which extends over East Asia and Australia, and opposing upper-level convergence over the Atlantic domain and neighboring continents (Supplementary Fig. 4j).

Overall, current trends in climate variables do not align well with CMIP6 historical simulations and future projections, which both typically indicate an El Niño-like SST pattern and a weakening of the atmospheric circulation consistent with a hybrid between the DeCO$_2$, WeGW, and WaGW theories[16,18,19,23,39]. These inconsistencies reinforce the idea that the present period may represent a transitional phase in the climate system, as proposed by Watanabe et al.[33] or that the response to anthropogenic forcing is masked by internal variability or is not emerging yet in observations[62] or, finally, that the response to the anthropogenic forcing in the models is muted by the model's biases[28,30]. In the following section, we demonstrate that the observed tropical rainfall trends are fully under the control of dynamical processes rather than thermodynamic ones. This suggests the key-role of

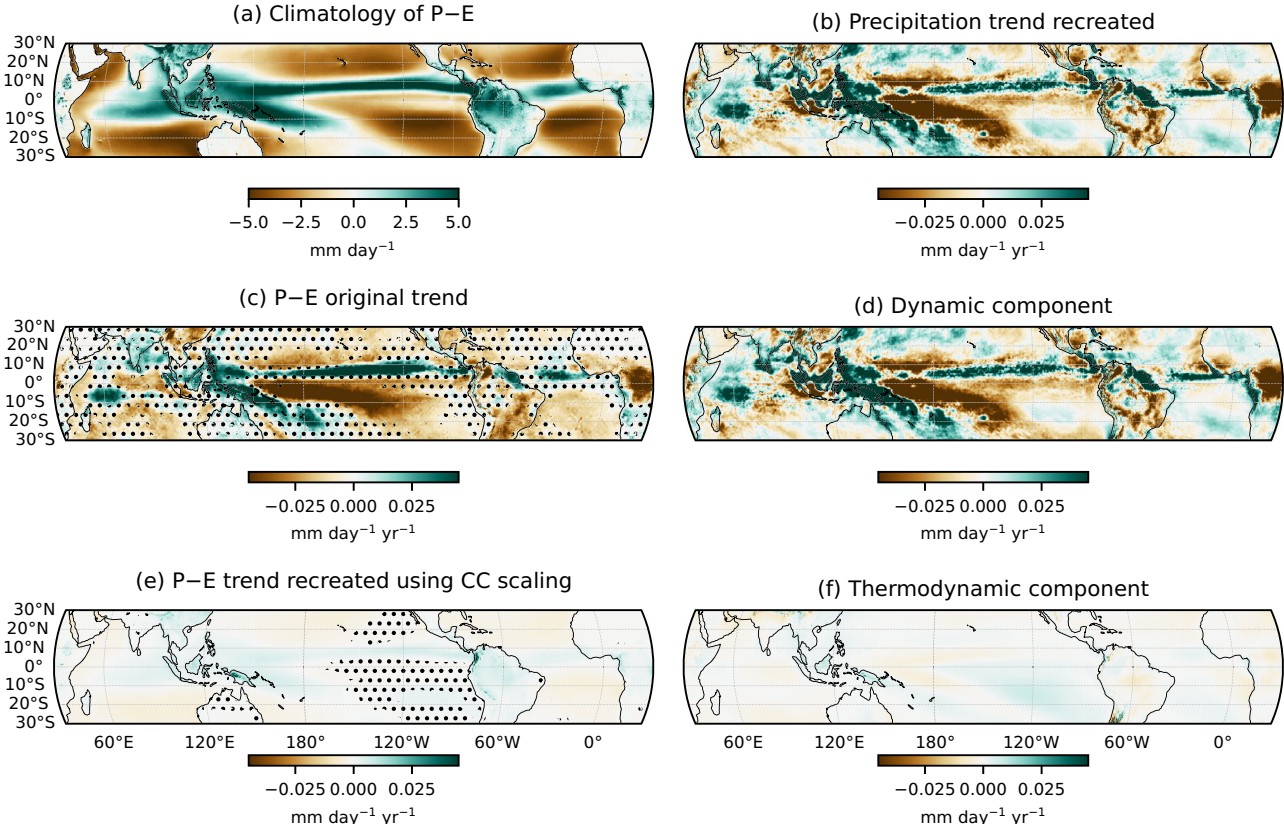

**Fig. 2 | Decomposition of tropical precipitation-minus-evaporation (P-E) and precipitation trends (1979-2024).** Panel (**a**) shows the climatology of P-E (mm day⁻¹). Panel (**b**) shows the recreated precipitation trend (mm day⁻¹ yr⁻¹) using dynamic and thermodynamic components using Equation (2). Panels (**c**) and (**e**) depict, respectively, the original and Clausius-Clapeyron reconstructed P-E trends (mm day⁻¹ yr⁻¹) computed using Equation (1), with stippling marking regions where the trends are not statistically significant at the 90% confidence level. Panels (**d**) and (**f**) show the dynamic and thermodynamic components of the precipitation trend (mm day⁻¹ yr⁻¹) computed using Equation (2). All trends are computed using ERA5 reanalysis for physical consistency.

surface temperature gradients in driving the atmospheric circulation, which will be examined next.

### Clausius-clapeyron scaling and dynamical processes

According to the Clausius-Clapeyron (CC) scaling, a warmer atmosphere can hold more moisture, implying that wet regions are expected to get wetter and dry regions drier under climate change, provided that relative humidity and vertically integrated moisture flux do not change[15,22]. In this framework, regions where precipitation exceeds evaporation ($P - E > 0$) are classified as wet, and regions with ($P - E < 0$) are considered dry. We apply CC scaling locally using Equation (1) to assess its ability to reproduce observed trends in P-E at each grid point.

Figure 2 (left column; panels a, c, and e) presents the climatology of $P - E$, the observed trend in P-E, and the trend estimated using CC scaling, respectively. While the spatial pattern of $P - E$ trends broadly resembles that of precipitation trends, notable differences exist. For example, over the Pacific Ocean, the drying band south of the equator is more spatially extensive in the P-E trend compared to the precipitation trend, while the wetting band north of the equator is narrower.

Under CC scaling, we expect the reconstructed $P - E$ trends to mirror the spatial structure of the $P - E$ climatology-that is, wet regions becoming wetter and dry regions drier. However, the reconstructed precipitation trends using the CC-scaling (Fig. 2e) exhibit very low magnitudes, roughly an order of magnitude smaller than the observed trends. This suggests that thermodynamic effects play only a limited role in explaining the observed trends in $P - E$.

To investigate this further, we employ a simplified two-layer moisture budget framework (Equation (2)) to decompose precipitation trends

into thermodynamic and dynamic contributions. The total reconstructed precipitation trend, along with the thermodynamic and dynamic components, is shown in Fig. 2 (right column; panels b, d, and f). The simplified moisture budget broadly reproduces the observed precipitation trends, with some regional differences, for instance, a narrower band of enhanced precipitation north of the equator. Crucially, the decomposition reveals that the dynamic component dominates the precipitation trend, while the thermodynamic contribution is comparatively very weak. This finding is consistent with the negligible trends obtained from the CC-based reconstruction of $P - E$, further confirming that present-day changes in precipitation and $P - E$ are primarily driven by changes in atmospheric dynamics, rather than thermodynamic constraints.

To assess the robustness of these results, we apply an alternative moisture budget decomposition framework adapted from Chadwick et al.[18], to reanalysis data. Details of this analysis are provided in Supplementary Section 3 and Supplementary Fig. 5. Consistent with the simple moisture budget decomposition, this independent framework also identifies dynamical changes (more precisely, spatial shifts in the pattern of convection centers, not the large-scale weakening of tropical circulation) as the dominant contributor to the observed precipitation changes, with thermodynamic contribution and upward vertical mass flux due to the weakening of the large-scale circulation playing only a secondary role. This demonstrates that our conclusions are not sensitive to the choice of decomposition framework.

Taking into account this important result, we now turn our attention to possible temperature drivers of both the observed trends of precipitation and atmospheric circulation in the next section.

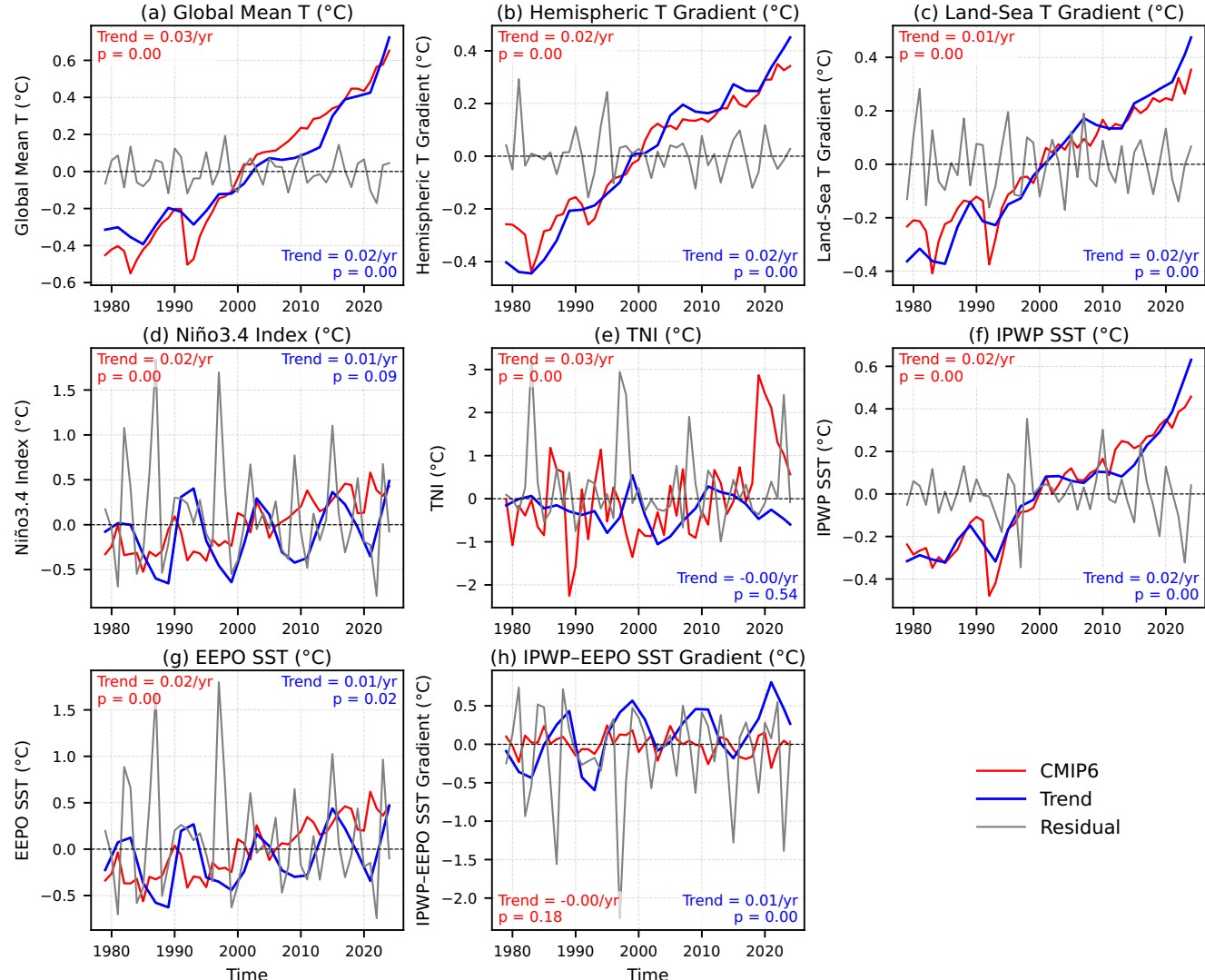

**Fig. 3 | Time series of eight climate indices from 1979 to 2024.** Each panel shows the trend (blue) and residual (gray) components, and the CMIP6 MME (red) estimate for a given index. The indices include: **a** Global Mean Temperature (GMT), **b** inter-hemispheric temperature gradient, **c** land-sea temperature gradient, **d** Niño 3.4 index, **e** Trans-Niño Index (TNI), **f** Indo-Pacific Warm Pool Sea Surface Temperature (IPWP SST), **g** Equatorial Eastern Pacific Sea Surface Temperature (EEPO SST), and **h** the IPWP-EEPO SST difference. The estimated linear trend (°C/year) and the corresponding p-value for the raw ERA5 (blue) and CMIP6 MME (red) time series of each index are also indicated in each panel. All trend and residual time series from ERA5 are calculated using LOESS, described in the Methods.

## Possible drivers of precipitation and atmospheric circulation trends

Since thermodynamic arguments alone do not adequately explain the observed precipitation trends, we apply a linear regression framework to assess the influence of various temperature-related indices on the precipitation and circulation trends. These include the Global Mean Temperature (GMT), inter-hemispheric and land-sea thermal gradients, Niño 3.4 index, Trans-Niño Index (TNI), Indo-Pacific Warm Pool SST (IPWP), Equatorial Eastern Pacific SST (EEPO), and their difference (IPWP-EEPO) as described in the Methods. To isolate long-term signals, we decompose each time series into a trend component and residual time series with the help of a LOESS smoother, as described in the Methods. Figure 3 presents the estimated trend and residual components of the annual time series of these eight indices derived from ERA5. For comparison, we also include the same indices derived from the CMIP6 MME, but without any filtering, to evaluate whether the models capture these observed variations.

As expected, GMT, the inter-hemispheric and land-sea thermal gradients, and IPWP SST all increase significantly, with the steepest rise in GMT, followed by IPWP SST and the two gradients. ERA5 and CMIP6

agree closely on these four indices, indicating robust greenhouse-gas (GHG)-forced trends. A cross-correlation analysis (Supplementary Fig. 6) further confirms strong agreement between ERA5 and CMIP6 for GMT (correlation = 0.97), the inter-hemispheric thermal gradient (0.98), the land-sea thermal gradient (0.97), and IPWP SST (0.96). Very high correlation (0.99) is also found between the inter-hemispheric and land-sea thermal gradients, both in ERA5 and CMIP6, indicating redundancy, consistent with the spatial patterns of warming in both ERA5 and CMIP6 (Fig. 1b and Supplementary Fig. 2e). Therefore, we exclude the inter-hemispheric thermal gradient in subsequent analyses, since the two thermal gradients are very highly correlated and yield similar results.

The Niño 3.4 index shows only a weak upward trend in ERA5 but a much stronger, highly significant rise in CMIP6, about three times larger, with decadal oscillations superposed in ERA5 but a more persistent increase in CMIP6 MME. The TNI declines insignificantly in ERA5, while CMIP6 shows an abrupt rise after 2020, likely due to outlier models. TNI shows very low correlation between ERA5 and CMIP6 (Supplementary Fig. 6). Therefore, due to these disagreements, we also exclude TNI in subsequent analyses.

By contrast, IPWP SST rises smoothly and significantly in both datasets, confirming a strengthened Indo-Pacific warm pool under GHG forcing. EEPO SST shows only a slight, variability-modulated rise in ERA5, but a sharp increase in CMIP6, paralleling Niño 3.4. Consequently, the IPWP-EEPO gradient trends upward in ERA5, consistent with recent findings[33], but slightly trends downward but insignificantly in CMIP6. This contrast suggests that the observed increase reflects the ocean thermostat mechanism[28,63], whereas the simulated decrease may result from damping processes such as evaporative cooling or reduced vertical mass flux[15,35,64].

To assess the role of temperature-related indices in driving precipitation and circulation trends, we regressed annual precipitation and 200-hPa velocity potential anomalies onto the trend components of the six selected indices in ERA5 (Fig. 4). The strongest signals, and best match to observed trends, are found in decreasing order for the land-sea thermal gradient, IPWP SST and GMT. Atmospheric regression patterns for the land-sea gradient and IPWP SST closely resemble the observed trends, with negative values (upper-level divergence) over the Indo-Pacific and positive values (convergence) over the southeast Pacific. By contrast, regressions with Niño 3.4 yield the expected Matsuno-Gill type response-positive precipitation anomalies over the central Pacific and negative over the Maritime Continent, with the associated velocity potential showing a zonal, symmetric structure. These patterns resemble El Niño/La Niña composites (Supplementary Fig. 3), but differ from the observed linear trends, which instead show wetting and upper-level divergence over the Indo-Pacific warm pool. This indicates that the Niño 3.4 trend component alone cannot explain the observed changes.

Interestingly, the equatorial Pacific SST gradient (IPWP-EEPO) shows an enhanced rainfall contrast between the Maritime Continent and the western Pacific south of the equator, but it fails to capture the wetting trend north of the equator as the land-sea thermal gradient and IPWP SST. The IPWP-EEPO regression with velocity potential resembles observed trends, with enhanced upper-level divergence over the Indo-Pacific and convergence over the eastern tropical Pacific. This supports the arguments that an increased Pacific SST gradient drives Walker circulation intensification under GHG forcing[32,34]. However, the regression misses the observed asymmetry: it shows broad convergence across the eastern Pacific, whereas observations feature a more localized center in the southeast Pacific. These results suggest IPWP SST changes exert a stronger and more direct influence on rainfall and circulation than the zonal SST gradient alone. The IPWP trend is also significantly correlated with thermal gradient indices (Supplementary Fig. 6), implying influence from broader climate-change signals such as the GMT and land-sea contrast, while also potentially feeding back onto them.

Repeating the regression analysis of ERA5 precipitation and upper velocity potential onto CMIP6 MME indices (Supplementary Fig. 7) reveals both similarities and important differences. As in ERA5, regressions with GMT, the land-sea gradient, and IPWP SST are consistent, reflecting the impact of GHG forcing on these trends. The land-sea gradient regression with velocity potential is particularly strong and best matches observations, suggesting a dominant role in driving circulation and rainfall changes. By contrast, regressions with Niño 3.4 and EEPO SST in CMIP6 show opposite signals to ERA5, likely due to their much stronger upward trends in CMIP6. Similarly, the equatorial Pacific SST gradient regression produces upper-level convergence over the Maritime Continent and near-zero values in the eastern Pacific, unlike ERA5. This discrepancy, consistent with the time series in Fig. 3h, indicates that CMIP6 historical simulations are overly sensitive to small equatorial Pacific SST gradient changes.

Repeating the analysis with residuals of the six indices obtained by removing their long-term trends (Supplementary Fig. 8) produced high local correlations, particularly with GMT and IPWP SST. However,

the spatial patterns differ from the observed precipitation and velocity potential trends (Fig. 1a, e), indicating that the residuals, which represent internal climate variability across different time scales, can modulate rainfall and circulation but do not explain their long-term changes.

Overall, our results highlight three correlated, climate-change-related indices, the inter-hemispheric and land-sea thermal gradients, and the intensification of the Indo-Pacific warm pool, as leading candidates driving the observed tropical precipitation trends. While land is often assumed to be passive relative to the ocean in the context of tropical variability, the following section shows that warmer land can causally influence tropical rainfall patterns, as demonstrated by coupled model sensitivity experiments with perturbations to land surface albedo.

## Sensitivity experiments: The role of the land–sea thermal gradient

To test the hypothesis that the land-sea thermal contrast can indeed alter atmospheric circulation and rainfall patterns, we performed global coupled simulations in which land surfaces were artificially warmed via surface albedo perturbations applied at global or regional scales. We used two global coupled models, the Climate Forecast System version 2 (CFS hereafter) and SINTEX-F2 (SINTEX hereafter), each implemented in two distinct configurations to test the robustness of the simulated responses. Importantly, both models exhibit good skill in simulating tropical variability (see Methods), despite sharing several biases common to CMIP models (see Supplementary Section 2 and Supplementary Fig. 9). The use of two independent coupled models and multiple configurations enhances confidence in the robustness and generality of the simulated circulation and rainfall responses. Details of the model formulations, configurations, and experimental design are provided in the Methods and Table 1. For clarity and due to space constraints, we present results from the SINTEX experiments here, while the corresponding results obtained with CFS are shown in the Supplementary Information.

In the first set of sensitivity experiments (zero-albedo), the land snow-free background albedo used in the land albedo schemes of both models was set to zero over all land areas. As a result, in the absence of snow, the land surface absorbs nearly all incident shortwave radiation (direct and diffuse) at each time step. These idealized zero-albedo experiments were compared with control simulations employing either the standard land albedo schemes (ctrl;[65,66]) or an updated configuration in which the model's snow-free background land albedo is replaced by satellite-driven MODIS albedo (MODIS-ctrl;[11]).

Figure 5 shows the differences between zero-albedo and MODIS-ctrl experiments for four variables: surface temperature, precipitation, 850-hPa winds, and 200-hPa velocity potential, based on the upgraded configuration of SINTEX (the same for CFS is shown in Supplementary Fig. 10). Supplementary Fig. 11 presents the corresponding differences (zero-albedo minus ctrl) for the standard configurations of both models and confirms the robustness of the response to the albedo perturbation, as the spatial patterns and amplitude of the anomalies are highly consistent between the standard and upgraded model configurations and between the two models.

As shown in Fig. 5a, the albedo perturbation warms land surfaces and produces global surface temperature changes that closely resemble observed trends (Fig. 1b,c). The surface warming over land emerges in a few days to weeks owing to the reduced heat capacity of land surfaces. The resulting temperature changes extend beyond land to the oceans, with particularly pronounced signals in the Pacific, including an enhanced equatorial SST gradient between the western and central Pacific, as well as cooling in the SH mid-latitudes. These oceanic responses emerge most clearly during boreal summer, consistent with the results of Terray et al.[11]. Despite the same albedo perturbations applied in both hemispheres, the warming is

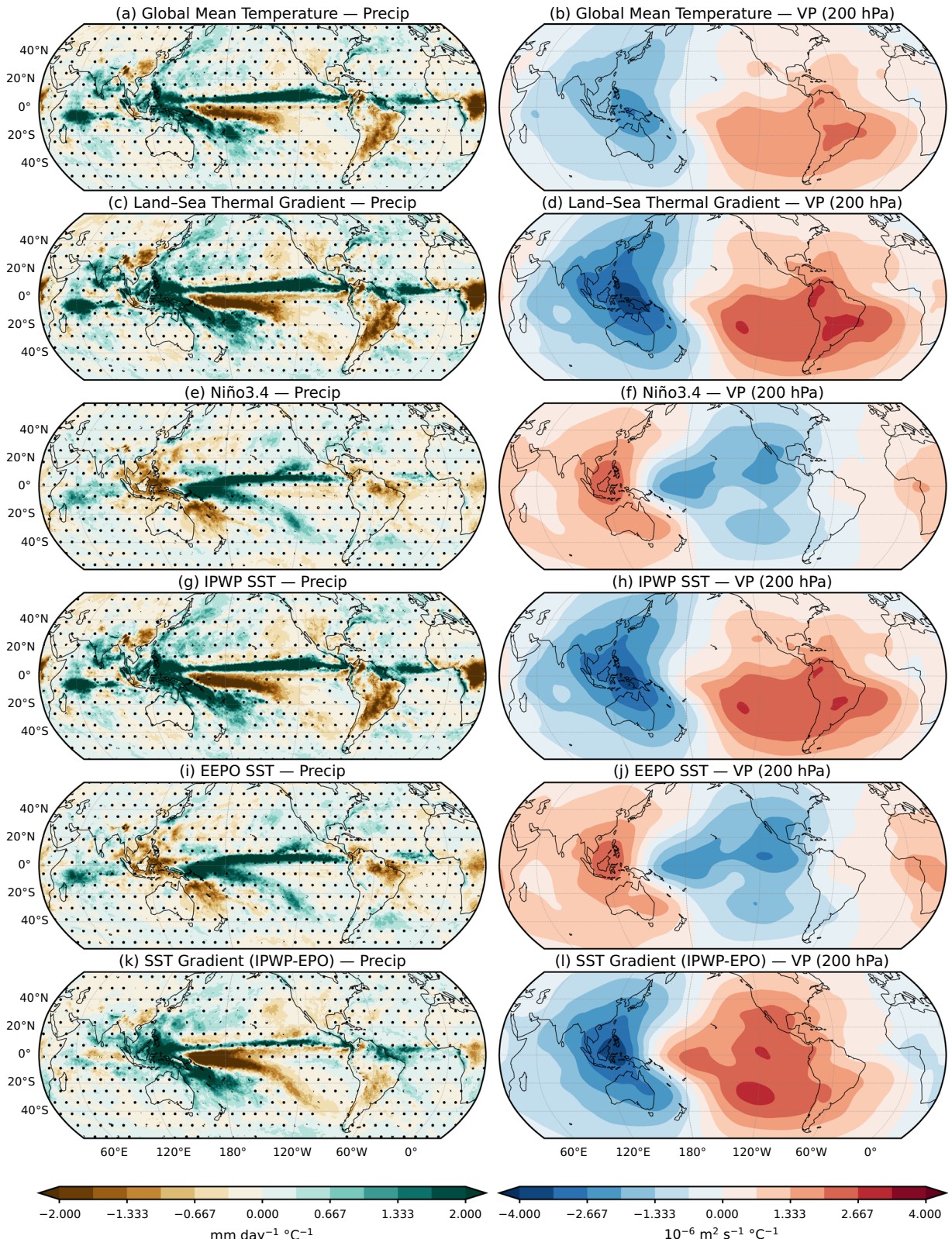

**Fig. 4 | Spatial regressions of annual precipitation and 200-hPa velocity potential anomalies (1979–2024) onto the trends of selected climate indices.** Left column: precipitation (mm day$^{-1}$ °C$^{-1}$); right column: 200-hPa velocity potential (10$^{-6}$ m$^2$ s$^{-1}$ °C$^{-1}$), both derived from ERA5. Fields are regressed onto the trend component of six temperature-related indices derived from the ERA5:**a, b** Global Mean Temperature (GMT), **c, d** land-sea thermal gradient, **e, f** Niño 3.4, **g, h** Indo-Pacific Warm Pool Sea Surface Temperature (IPWP SST), **i, j** Equatorial Eastern Pacific Sea Surface Temperature (EEPO SST), and **k, l** equatorial Pacific SST gradient (IPWP-EEPO). Colors show regression coefficients; stippling marks grid points where the coefficient is not significant at 90% confidence level.

**Table 1 | Summary of coupled model control and sensitivity experiments**

| Experiment | Land albedo configuration | CFS | SINTEX |
|---|---|---|---|
| ctrl | Standard prescribed snow-free background land albedo | 80 | 210 |
| MODIS-ctrl | MODIS-based snow-free background land albedo | 60 | 110 |
| zero-albedo | Snow-free background land albedo set to zero globally | 30 | 60 |
| desert-albedo | MODIS background retained; snow-free land albedo additionally reduced by 20% over the Sahara–Arabian–Pakistan–Thar desert region (15°–40°N, 20°W–75°E) | 30 | 60 |

All simulations use fixed present-day atmospheric $CO_2$ concentrations. Integration length is shown in years.

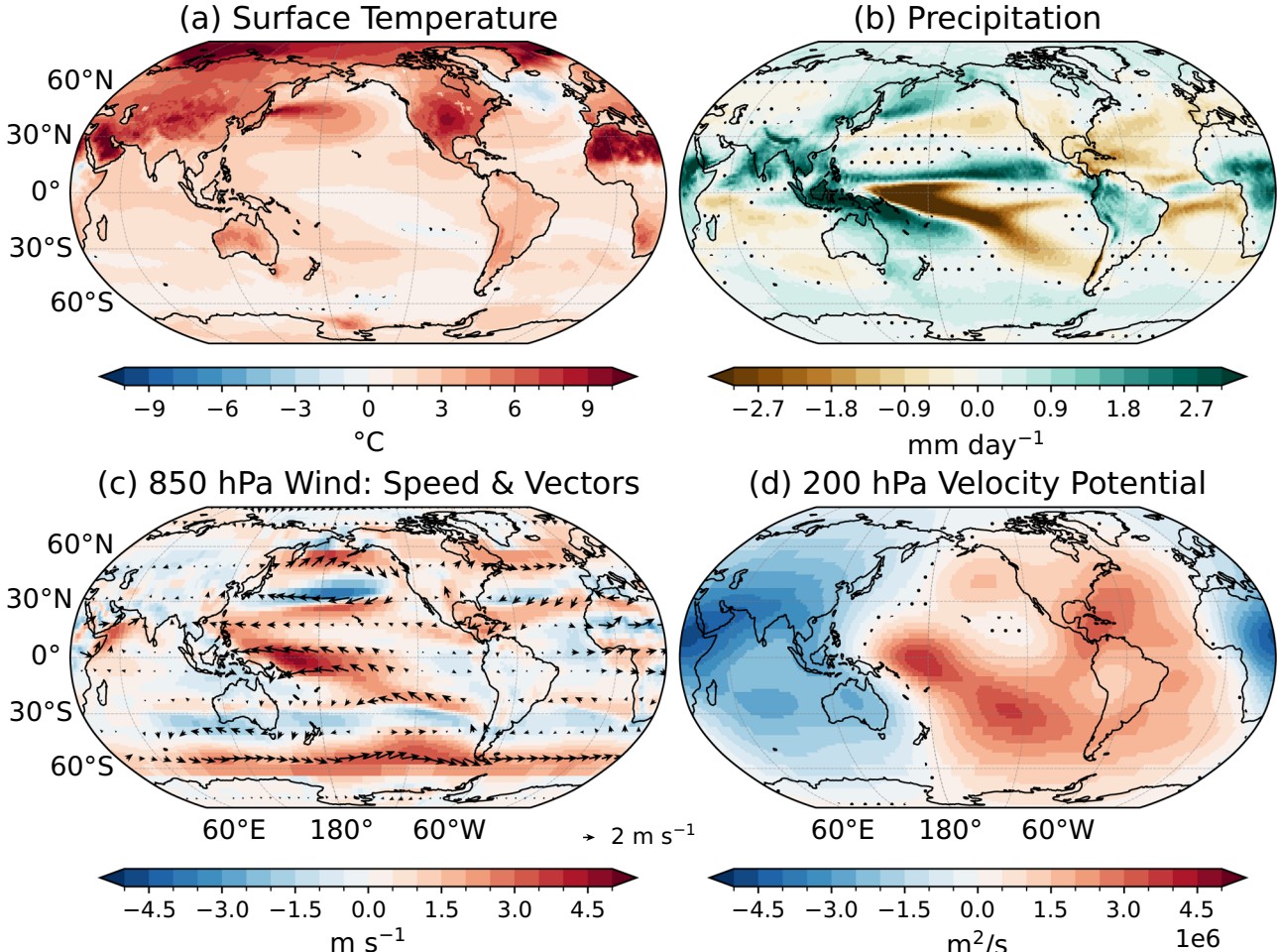

**Fig. 5 | Climate model sensitivity to enhanced land-sea thermal contrast.**
Results are from a coupled experiment using the SINTEX model in which the snow-free background land albedo is set to zero, maximizing land surface warming relative to the oceans. Panels show the difference between the sensitivity experiments and the MODIS-ctrl simulation for **a** surface temperature (°C), **b** precipitation (mm/day), **c** 850 hPa wind speed (m s$^{-1}$) with vectors indicating wind direction, and **d** 200 hPa velocity potential (m$^2$ s$^{-1}$). Dots represent regions with statistically insignificant differences at 90% confidence level according to the Welch's two-sample t-test. The wind vectors in panel (**c**) are plotted only if at least one component of the wind difference is statistically significant at 90% confidence level.

substantially stronger in the NH, indicating the existence of positive feedbacks that amplify the initial forcing. This suggests that the land-sea and hemispheric thermal contrasts are not merely transient responses associated with differing land-ocean heat capacities but instead reflect complex feedback processes involving evaporation, rainfall, moisture, snow, and cloud feedbacks, consistent with previous findings from CMIP simulations and projections[52–55,67].

Importantly, the enhanced NH land warming closely resembles recent observations (Fig. 1b), especially at high latitudes and over the subtropical arid regions such as the Sahara, Arabia, and the Middle East. These features are consistent with the observed polar (or high

latitude) and desert amplification patterns that characterize recent climate change[49,50,68,69], and with the strong correlation between land-sea and hemispheric thermal contrasts seen in both ERA5 and CMIP simulations (Supplementary Fig. 6). The SST response further exhibits a La Niña-like pattern (Fig. 5a), suggesting that the observed cooling trend in the Pacific (Fig. 1c) may be partly driven by forced changes in the land-sea thermal gradient, rather than arising solely from internal variability[36,37,58,62].

These temperature patterns induce a dynamical response: stronger NH land warming at high and subtropical latitudes enhances both meridional and zonal thermal gradients, leading to large-scale

adjustments in surface pressure fields[40]. The resulting circulation changes are evident at both surface and upper levels (Fig. 5c, d). At the surface, the response includes markedly stronger southeast trade winds in the South Pacific and intensified SH mid-latitude westerlies, consistent with the relative cooling in these regions. Both models also show strengthened subtropical anticyclones (as seen from 850-hPa wind anomalies), particularly in the SH, further reinforcing cooling over the trade-wind regions. These differences closely resemble observed SST and surface wind trend patterns, especially over the Pacific (Fig. 1b, c). As a quantitative measure, the WCI index (defined in Global Climate Trends) increases by 217% and 117% relative to MODIS-ctrl and by 192% and 112% relative to ctrl, in the CFS and SINTEX models, respectively. This response is consistent with the significant positive linear trend in the WCI in ERA5 as noted in Global Climate Trends[33].

The precipitation differences (Fig. 5b) also show strong similarities to the observed precipitation trends (Fig. 1a), particularly the wetting north of the equator, over the western Pacific, the northern Indian region, and the drying south of the equatorial Pacific. However, some regional discrepancies remain, for example, the precipitation trend shows drying over central Africa in ERA5, whereas the sensitivity experiments yield strong positive anomalies in this region. This mismatch likely reflects additional processes not captured by the land-sea thermal gradient mechanism alone, such as the intensification of the Indo-Pacific warm pool under $CO_2$ radiative forcing. However, the observed trend over central Africa is itself uncertain, as this region shows substantial disagreement between ERA5 and GPCP (Fig. 1a; Supplementary Fig. 1a). Rainfall trends over the tropical Atlantic also differ markedly between observations/reanalyses and the idealized zero-albedo experiments.

Despite these limitations, the experiments provide insight into the mechanisms behind the observed tropical Indo-Pacific rainfall trends over both land and ocean. We therefore next examine the processes responsible for the simulated tropical circulation and rainfall responses to land surface albedo perturbations by analyzing energy-budget and albedo diagnostics at both the surface and the top of the atmosphere (TOA), as shown in Fig. 6 (the same for CFS is shown in Supplementary Fig. 12).

As expected, surface albedo decreases over all land areas in both hemispheres in the zero-albedo experiments; however, the reduction is substantially larger over NH subtropical arid regions, particularly the Sahara, Arabia, and the Middle East (Fig. 6a). Consistent with this enhanced albedo reduction, net surface shortwave radiation increases significantly over these arid regions, with the excess energy largely balanced by stronger upward longwave radiation and enhanced sensible heat fluxes from the surface (Fig. 6c, e, g). By contrast, and despite the imposed surface albedo perturbation, net surface shortwave radiation exhibits little change or even decreases over the neighboring African and Indian monsoon regions. This response is consistent with strong cloud and water vapor masking associated with deep convection. In these monsoon regions, surface net longwave radiation increases in parallel with enhanced rainfall, consistent with deeper convection and strong interactions between heated subtropical deserts and NH monsoon systems triggered by the albedo perturbation[11,70]. Over the tropical Pacific and Indian Oceans, the surface net shortwave and longwave radiations are also fully consistent with rainfall and SST responses in the zero-albedo experiments. These patterns can be explained by the cloud and water vapor masking effects combined with the SST-driven feedbacks arising from the circulation response.

At the TOA, Earth's albedo also decreases over land, especially over arid regions, in the idealized zero-albedo experiments (Fig. 6b). Consistently, net shortwave radiation is again maximum over the subtropical arid regions of the NH and bears a strong similarity with the net shortwave budget at the surface (Fig. 6c). As expected, OLR

increases in all regions where cloud masking is absent in the zero-albedo experiments, including the NH subtropical deserts (Fig. 6f). Despite this enhanced OLR, the TOA net radiation is maximum over these arid areas, demonstrating that the net atmospheric energy input is maximum there in the zero-albedo runs (Fig. 6h). Interestingly, direct radiative forcing over land (e.g. increased CO2 with fixed SST) is also characterized by an enhanced atmospheric energy input over the NH deserts, despite the fundamentally different physical mechanisms involved compared to the surface albedo perturbations applied here[20].

In a nutshell, the TOA radiation budget in the zero-albedo simulations indicates a strong increase in net atmospheric energy input over NH deserts, despite enhanced OLR. This enhanced energy input enables the "hot" deserts to interact with the NH monsoons, triggering the dynamical and rainfall responses and the Indo-Pacific teleconnection described above. To more directly attribute these circulation and rainfall responses to the net atmospheric energy input over the NH arid regions, we conducted additional sensitivity experiments (desert-albedo) using both CFS and SINTEX in their MODIS-ctrl configuration. In these experiments, the snow-free background land albedo was reduced by 20% only over the Sahara, Arabia, and Middle East arid regions ((land regions over 15°–40°N and 20°W–75°E), Supplementary Figs. 13–15).

Importantly, the results show statistically significant precipitation and low- and upper-level circulation responses that are qualitatively similar to those obtained in the idealized zero-albedo experiments, but with substantially weaker magnitudes, as expected, and in closer agreement with the observed rainfall and circulation trends (e.g. compare Figs. 1 and 5; Supplementary Fig. 13). This supports the hypothesis that the enhanced land-sea contrast[54] and ongoing desert amplification[68] play significant roles in shaping the observed tropical precipitation trends and further highlights the importance of warming over the major desert regions of the NH in driving the associated rainfall and circulation responses. Consistent with this interpretation, the net atmospheric energy input in the desert-albedo experiments is confined to these arid regions. Although the mechanisms responsible for surface warming in these idealized experiments differ from those operating in the real climate system[20,68], the experiments nevertheless provide a useful framework for isolating the dynamical response to desert warming and the associated increase in surface sensible heat flux, as observed in the real climate.

Dynamically, enhanced warming over NH land and arid regions strengthens subtropical anticyclones and trade winds, leading to cooling of the underlying ocean. As a result, warmer NH land leads to a relatively cooler SST over the tropical SH, especially over the Pacific. Furthermore, the stronger atmospheric energy input in the NH (Fig. 6 and Supplementary Figs. 12, 14, and 15) shifts tropical precipitation northward in all idealized experiments. To provide a quantitative view of this northward shift of the ITCZ in the experiments, we use a Precipitation Asymmetric Index (PAI) with respect to the equator, following Hwang and Frierson[71]. The PAI is defined as the difference between area-averaged precipitation in the NH tropics (equator-20°N) and SH tropics (equator-20°S), normalized by the mean tropical precipitation (20°S-20°N), and provides a measure of the fraction of ITCZ rainfall occurring in the NH. At the global scale, the PAI increases by 86% and 32% in the zero-albedo and desert-albedo experiments relative to MODIS-ctrl in CFS, and by 41% and 13% in SINTEX, respectively. These results are consistent with energetic theories predicting that the ITCZ migrates toward the hemisphere receiving greater energy input[72]. Although the excess energy is exported to the opposite hemisphere through the atmosphere or ocean, in the tropics, this compensation is dominated by inter-hemispheric ocean heat transport[73], highlighting the importance of dynamical adjustments. The northward rainfall shift is most pronounced over the tropical Pacific, but also occurs over the Indian subcontinent and West Africa, where monsoon rainfall intensifies[11,70].

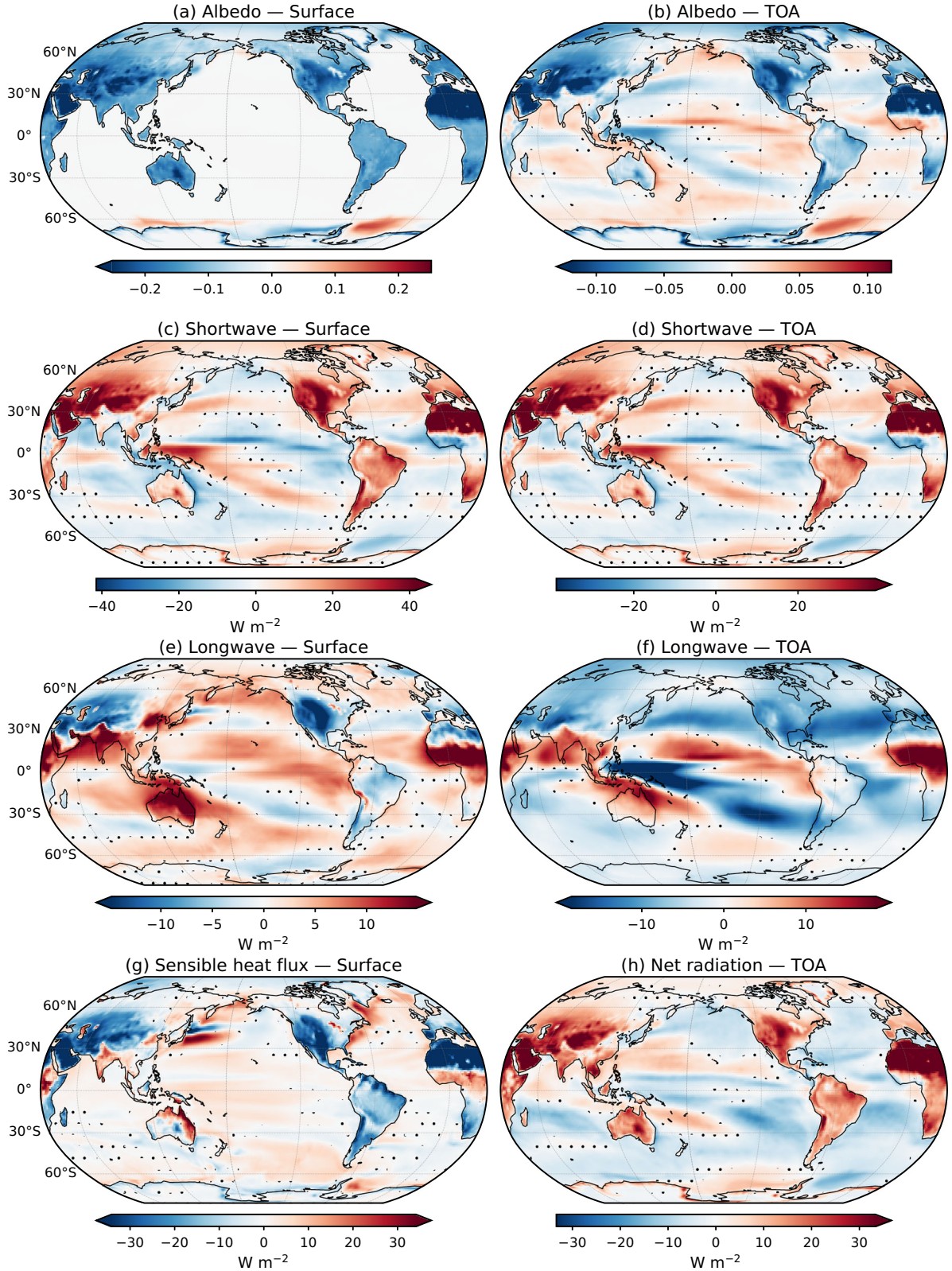

**Fig. 6 | Energy flux response to enhanced land-sea thermal contrast.** Spatial patterns of the response to an imposed increase in land-sea thermal contrast, obtained from coupled SINTEX sensitivity experiments in which the snow-free background land albedo is set to zero. Shown are the differences between the sensitivity experiment and the MODIS-ctrl simulation. Left panels show surface responses, and right panels show Top of the Atmosphere (TOA) responses for **a**, **b** albedo, **c**, **d** shortwave radiation, **e**, **f** longwave radiation, **g** sensible heat flux at the surface, and **h** net TOA radiation (units: W m$^{-2}$ for radiation and sensible heat fluxes). Note that downward (upward) fluxes are positive (negative). Dots indicate regions where differences are not statistically significant at the 90% confidence level based on Welch's two-sample t-test.

Both of these monsoon systems are adjacent to the NH sub-tropical deserts, over which the albedo perturbations are imposed in the simulations, and the net atmospheric inputs are at a maximum. Enhanced sensible heat flux from these deserts warms the lower troposphere, destabilizes the atmosphere at the margins of the monsoon domains, and triggers substantial rainfall increase over both West Africa and India (Fig. 5, Supplementary Fig. 10,[11,70]). This mechanism contributes to the strengthened hemispheric asymmetry of tropical rainfall in the experiments, as reflected by the increased global PAI (see also Fig. 5c,d). Importantly, observed rainfall trends over recent decades exhibit a comparable asymmetry, with a strengthened North ITCZ over the Indo-Pacific (Fig. 1a), as the PAI linear trends estimated from ERA5 precipitation are also positive and significant at 90% confidence level at both the global scale and in the Pacific (0.00157/yr and 0.00224/yr, respectively). This indicates that tropical rainfall trends are not governed solely by changes in the zonal Walker circulation.

The Indo-Pacific precipitation, SST, and atmospheric responses to albedo perturbation closely resemble observed trends in these areas (Fig. 1a,b), supporting our hypothesis that warmer NH land and arid regions under anthropogenic forcing (Fig. 5) can induce similar rainfall changes by altering large-scale circulation. However, the 200-hPa velocity potential patterns in the zero-albedo and desert-albedo experiments are more asymmetric than in ERA5 (Fig. 1e), implying that NH land warming alone cannot explain all the observed Walker circulation changes.

This suggests that observed circulation trends arise from interacting mechanisms, including desertification, enhanced land-sea contrast, Indo-Pacific warm pool intensification, Pacific SST gradients, and internal variability, as highlighted in previous studies.

## Discussion

In this study, we investigate the tropical rainfall trends during the satellite era to assess our understanding of ongoing rainfall regime shifts in some of the world's most populated regions. Observations and reanalyses reveal several robust signs: intensified rainfall in the northern ITCZ over the Pacific and Atlantic, the SPCZ (also more tilted), the Maritime Continent, and India, contrasted with drying in the SH, particularly over the South Pacific and South America. By contrast, CMIP6 historical simulations fail to reproduce these features, instead producing an emerging El Niño-like rainfall pattern that closely resembles (though with weaker intensity) the far-future rainfall changes these models project for the 21$^{st}$ century.

These results suggest three possible, but not exclusive, scenarios: (i) the observed rainfall trends are governed by decadal natural variability rather than by current radiative forcing; (ii), they represent only a transient response to this radiative forcing; or (iii) they expose a major failure of current climate models to simulate the tropical rainfall response to anthropogenic forcing. This debate mirrors exactly the ongoing discussion in the literature about the "tug of war" between different oceanic and atmospheric mechanisms, such as evaporative damping, reduction in the vertical mass flux linked to atmospheric energetics, the oceanic tunnel, and the ocean thermostat (see Heede et al.[35] for an excellent review and explanation of these different mechanisms). These competing explanations have been proposed to account for the contrasting behavior of the Pacific Walker circulation observed during the recent decades with that simulated or projected by successive generations of CMIP models over the past two decades[27–32,35,62,74].

To assess the likelihood of these alternate scenarios, we first demonstrate that these trends are driven by dynamical precipitation changes associated with three large-scale surface temperature drivers: the intensification (and extension, as described in Weller et al[57]) of the Indo-Pacific warm pool, the inter-hemispheric and land-sea thermal contrasts at the global scale. Furthermore, it is shown that the latest generation of models reproduced fairly well the observed trends of these three large-scale drivers; however, they still fail to simulate either the observed rainfall pattern or the associated strengthening of the Walker circulation during the satellite era correctly. The land-sea and inter-hemispheric warming contrasts are tightly coupled and robust features of human-induced global warming. They are not merely a transient effect of global warming associated with the differences in heat capacity and ocean heat uptake between land and ocean, but also arise from distinct climate feedbacks operating under transient, near-equilibrium, and equilibrium conditions[54,55].

The intensification of the Indo-Pacific warm pool is a forced response to radiative forcing, probably driven by other processes like the ocean thermostat mechanism, which indicates that the Pacific equatorial SST gradient may be strengthened by the local Bjerknes feedback in response to global warming[63]. Our analysis shows that this warm pool intensification across both the Indian and Pacific oceans is physically consistent with the observed strengthening and westward-shifting trend of the Walker circulation[75], which dominates during boreal winter (Supplementary Fig. 4j). This intensification is as important a driver of the tropical rainfall changes under global warming as the thermal contrasts. Furthermore, both factors are statistically correlated and show comparable skill at reproducing the observed rainfall trends, suggesting that warm-pool intensification alone could explain the observed rainfall response.

However, dedicated coupled sensitivity experiments with albedo perturbations rule out this possibility, showing a clear cause-and-effect link. Enhanced NH warming, land-sea warming contrasts, and desert amplification in the NH are initially driven by a rapid land warming response in the NH, owing to the reduced heat capacity of land surfaces. This rapid land warming subsequently induces cooling of SSTs in the southeast Pacific and the SH as a dynamical response during boreal summer. Together, these processes reproduce both the observed warming pattern and associated rainfall and circulation changes in the tropical Indo-Pacific regions. This leads to the hypothesis that both the land-sea warming contrast and the warm pool intensification act together to constructively shape recent observed rainfall and circulation trends. This interpretation is consistent with the modified ocean thermostat mechanism proposed by Seager et al.[28] in which enhanced inter-hemispheric and land-sea warming contrasts intensify the southeast Pacific trade winds, driving a strong evaporative cooling, which further amplifies local wind anomalies and, subsequently, triggers the ocean thermostat mechanism along the equatorial Pacific. Since the land-ocean warming contrast and desert amplification are robust and persistent features associated with GHG induced warming[54,68], this scenario challenges the idea that warm pool intensification, the ocean thermostat mechanism, and the associated strengthening of the Walker circulation are merely transient responses to the anthropogenic warming as suggested by idealized coupled experiments forced with a wide range of both abrupt and gradual $CO_2$ increases[35]. Furthermore, while previous studies focus mainly on atmosphere-ocean interactions, our results reveal that land warming plays an important role in shaping tropical rainfall and atmospheric responses under anthropogenic forcing and cannot be considered as a passive element of the climate system, as often assumed.

The land-sea thermal and interhemispheric thermal contrasts, however, are strongly coupled in the climate system. These two large-scale gradients are almost perfectly correlated in both ERA5 and CMIP6, reflecting the dominant land fraction in the NH. This near-degeneracy implies that processes that enhance the land-sea warming contrast inevitably modify the inter-hemispheric thermal gradient, and vice versa, making it difficult to fully disentangle their individual influences using idealized perturbation experiments. The results presented here should therefore be interpreted in terms of their combined effect, with land warming acting as an active component of the coupled atmosphere-ocean system that shapes tropical circulation and rainfall responses. In this framework, the intensification of the Indo-

Pacific warm pool also does not operate in isolation but interacts constructively with these large-scale thermal gradients to amplify regional precipitation and circulation changes. While isolating warm-pool-only effects would require a dedicated experimental design targeting regional oceanic perturbations, the present results emphasize that robust global-scale gradients provide the background state upon which the Indo-Pacific warm-pool dynamics act, together shaping the observed tropical climate trends during the satellite era.

Despite the framework's physical consistency, a critical question remains: what ultimately drives the discrepancies between observed rainfall and atmospheric trends, and those simulated by CMIP6 models, during the satellite era? First, consistent with previous studies, our analysis shows that although the CMIP6 models reproduce very well the trends in the three large-scale temperature drivers discussed above, they simulate a stronger warming in the equatorial cold tongue upwelling region and a reduced east-west SST gradient in the Pacific. As a consequence, the models produce a weakened Walker circulation and, ultimately, an El Niño-like rainfall pattern in their historical simulations, closely resembling their centennial response to radiative forcing. These biases suggest that oceanic and atmospheric mechanisms, such as evaporative damping, reduction in vertical mass flux, and the ocean tunnel, which favor a reduced equatorial SST gradient and weakened Walker circulation in the Pacific[35], already dominate the simulated historical response to radiative forcing, in stark contrast to what is observed.

Finally, we suggest that this inability of climate models to reproduce the observed changes may be partly associated with the failure of many atmospheric models to reproduce the asymmetric features of atmospheric and rainfall responses to the two large-scale thermal contrasts discussed above, as well as their positive interactions with the warm pool dynamic, as illustrated in Supplementary Fig. 7. Obviously, in CMIP6 models, this asymmetric atmospheric response to the land-sea thermal contrast is muted by the severe double-ITCZ and cold tongue biases (see[12,48,71] and Supplementary Fig. 2a,b for illustration), which instead promote rainfall intensification on both sides of the equator and prevent any westward extension of the drying tendency in the South Pacific seen in observations. The large and persistent dry and wet biases over, respectively, land and ocean over NH monsoon areas in coupled simulations[11], as well as coupled biases in the equatorial Indian ocean[10], may further play a role and enhance the zonal features of the forced tropical rainfall and atmospheric responses to radiative forcing in current coupled simulations and projections. It seems unlikely in this context that the current discrepancies between observations and CMIP6 historical simulations are simply the result of the natural decadal variability of the true climate system or of the inability of current climate models to simulate realistically such internal variability. Instead, our findings point to the fact that current climate models fail to simulate important aspects of forced tropical climate changes due to model's deficiencies, consistent with the views of refs. 28–30.

## Methods
### Data used
The monthly data sets for evaporation (E), precipitation (P), vertical pressure velocity, t2m, specific humidity, winds and SST are obtained from European Centre for Medium-Range Weather Forecasts Reanalysis 5 (ERA5) reanalysis[76]. The 200-hPa velocity potential fields are computed from the 200-hPa winds using the Windspharm Python package[77]. Previous studies have shown that the water budget in ERA5 is not fully closed[78]. However, it is the best available reanalysis product to date, providing all the necessary variables for detailed analysis. Furthermore, we have compared the results from ERA5 with those of other reanalysis or observation products, whose details are provided in the supplementary materials, and they are broadly consistent with those derived from ERA5. All analyses in this study are conducted for the period 1979–2024, corresponding to the satellite era. Monthly CMIP6 data sets for precipitation, t2m, and SST are also used. The CMIP6 historical simulations (1979–2014) are concatenated with the Shared Socioeconomic Pathway 2-4.5 (SSP2-4.5) to construct a continuous record for 1979-2024, enabling comparison with observations and reanalyses, since SSP2-4.5 is widely regarded as the most plausible near-term scenario[1]. A list of the CMIP6 models used for computing the MME is provided in Supplementary Table 1. Furthermore, the precipitation from SSP5-8.5 for 2081-2100 from the same set of climate models is used to analyse the future changes in the precipitation as displayed in Supplementary Fig. 2c.

### Clausius–clapeyron scaling
Assuming that climate change does not alter the large-scale integrated moisture flux in the atmosphere, relative humidity, or horizontal temperature gradients, the response of precipitation minus evaporation ($P - E$) can be approximated by a simple scaling based on the Clausius–Clapeyron (C–C) relationship, as shown in Equation (1)[15]:

$$\delta(P - E) \approx \alpha \, \delta T_s \, (P - E) \tag{1}$$

Here, $\delta(P - E)$ represents the trend in $P - E$ over time, and $\delta T_s$ is the anomaly in near-surface (2-meter) air temperature. The term $(P - E)$ is the long-term climatological mean of precipitation minus evaporation. This formulation assumes that the moisture increase in the atmosphere with warming follows the Clausius–Clapeyron relation.

The scaling coefficient $\alpha$ is defined as:

$$\alpha = \frac{L R_v}{T_s^2}$$

where:
- $L = 2.268 \times 10^6$ J/kg is the latent heat of vaporization,
- $R_v = 461.5$ J/(kg · K) is the gas constant for water vapor,
- $T_s$ is the climatological mean of near-surface air temperature at each grid point (in Kelvin).

Therefore, Equation (1) implies that the trend in $P - E$ is proportional to the surface temperature anomaly, scaled by $\alpha$ and the climatological values of $P - E$.

While this scaling has been widely applied in climate model projections at global or zonal scales[15,22], to the best of our knowledge, it has not yet been evaluated using historical observational data at a regional scale. This study aims to test the validity of this scaling in observed datasets and reanalyses over the tropics.

### Simplified moisture budget analyses
Changes in precipitation are driven by both dynamic and thermodynamic factors. Dynamic changes refer to shifts in intensity or atmospheric circulation patterns, while thermodynamic changes are primarily associated with the increased moisture-holding capacity of the atmosphere under warming conditions through the CC relationship. To isolate the contributions from these components to the observed precipitation trend, we first use a simplified two-layer moisture budget analysis technique[15,17,19,79]:

$$\Delta P \sim \frac{\Delta w \cdot \overline{q}}{g} + \frac{\overline{w} \cdot \Delta q}{g} \tag{2}$$

where $P$, $g$, and $q$ denote, respectively, precipitation (mm day$^{-1}$), the Earth's gravity (m s$^{-2}$), and near-surface specific humidity (e.g., at 1000-hPa; kg kg$^{-1}$), and $w$ is the pressure velocity at 500-hPa (Pa s$^{-1}$). $\overline{q}$ and $\overline{w}$ are, respectively, the climatological means of near-surface specific humidity and pressure velocity at 500-hPa during the 1979-2024 period. $\Delta P$, $\Delta w$, and $\Delta q$ are, respectively, the anomalies or trends in

precipitation, pressure velocity, and specific humidity relative to their climatological means. The first term, $\Delta w \cdot \bar{q}/g$, represents the dynamic contribution, capturing how changes in atmospheric circulation influence precipitation. The second term, $\bar{w} \cdot \Delta q/g$, represents the thermodynamic contribution, reflecting how changes in specific humidity affect precipitation. The scaling by $g$ in both terms allows them to be expressed in the same units as precipitation (e.g., mm/day), enabling a more quantitative comparison between $\Delta P$ and its dynamic and thermodynamic components. This simplified moisture budget can be derived from the vertically integrated moisture conservation equation by assuming a two-layer troposphere with a separation at 500-hPa and neglecting upper-tropospheric specific humidity. This two-layer approximation performs well in the tropics, particularly in the context of climate change[17,19,79].

To confirm the results, e.g., that the recent rainfall trends are almost exclusively driven by the dynamical contribution and shifts in convection patterns, we also used a completely different moisture decomposition developed in Chadwick et al.[18], which is detailed in Section 3 of the Supplementary Information.

## Definition of climate indices used

To investigate large-scale drivers of precipitation change, we computed a suite of climate indices based on near-surface air temperature (2-meter temperature, t2m) and SST anomaly fields. Anomalies were computed with respect to the climatology over the full analysis period (1979-2024). All spatial averages were computed using cosine-latitude weights to account for the varying surface area of grid cells with latitude. The following indices were constructed:

- **Global Mean Temperature:** The global mean temperature (GMT) anomaly was computed by applying an area-weighted average to the t2m field over all grid cells, using the cosine of latitude as weights.
- **Inter-Hemispheric Thermal Gradient:** This index is defined as the Northern Hemisphere mean t2m minus the Southern Hemisphere mean t2m (NH - SH), where each hemisphere's value is computed as an area-weighted average over 0°-90°.
- **Land–Sea Thermal Gradient:** Calculated as the difference between the weighted mean t2m over land grid points and that over ocean grid points.
- **Niño 3.4 Index:** The Niño 3.4 index represents the weighted area-averaged SST anomaly over the central equatorial Pacific region (5°S-5°N, 170°W-120°W) and is a standard ENSO index[80].
- **Trans-Niño Index:** The Trans-Niño Index was calculated as the standardized difference between SST anomalies in the Niño1+2 region (10°S-0°, 90°-80°W) and the Niño4 region (5°S-5°N, 160°E-210°E) following Trenberth and Stepaniak[81].
- **Indo-Pacific Warm Pool SST** This index was calculated as the average SST anomaly along the Indo-Pacific region (5°S-5°N, 80°E-150°E).
- **Equatorial East Pacific SST** This index was calculated as the average SST anomaly along the equatorial east Pacific region (5°S-5°N, 180°-80°W).
- **Equatorial Pacific SST Gradient:** Calculated as the zonal SST contrast along the equator and is defined as the SST difference between the Indo-Pacific Warm Pool SST (5°S-5°N, 80°E-150°E) and Equatorial East Pacific SST (5°S-5°N, 180°-80°W) following Watanabe et al.[33].

## Filtering the time series into trend and residuals

Observed or simulated climate trends are often estimated by fitting a straight line using linear least squares. However, climate trends-especially in the context of global warming-are often nonlinear, and more accurate methods than simple linear least squares fitting should be preferred.

Locally weighted regression, or LOESS, is a non-parametric method for fitting a smoothed regression curve to data through local smoothing. The method is nonlinear but does not require specifying the shape of the interpolating functions, unlike other nonlinear techniques such as exponential smoothing. Suppose $x_i$ and $y_i$ for $i = 1$ to $n$ are, respectively, measurements of one independent and one dependent variable. In the context of climate trend estimation, the independent variable $x_i$ is typically just the time index (e.g., $x_i = i$). LOESS provides a smoothed local estimate $g(x)$ of $y$ based on $x$[82,83]. More precisely, in the context of trend estimation, the LOESS method approximates the value of the trend at each point $y_i$ in a time series by fitting a polynomial of degree 1 or 2 using weighted least squares. This local polynomial regression is weighted by the distance between the point $x_i$ (for which the estimate is being made) and the surrounding points $x_j$.

The method has two main degrees of freedom. The first is the degree $d$ of the fitted polynomial, which controls the curvature of the interpolation and is typically chosen as 0, 1, or 2[83]. The second is a positive integer $q$ that defines the width of the moving window used for local fitting. As $q$ increases, the estimated trend $g(x)$ becomes smoother. In the limit as $q \to \infty$, the LOESS estimate converges to an ordinary least squares polynomial fit of degree $d$ applied to the full time series. Other important features of LOESS is the ability to produce robust estimates of the trend component that are not distorted by outliers or extreme values and the stationarity of the residual time series[83]. Here, we used the values 1 and 6 years for $d$ and $q$, respectively, and applied LOESS to yearly and monthly time series. However, as the results are identical and robust, we show here only the yearly results. Furthermore, a 4-year value for the $q$ parameter gives almost identical results.

## Trend and statistical significance tests

The trends are also calculated using the ordinary least squares linear regression method. The statistical significance of the trend is then assessed using the two-tailed t-test, with the null hypothesis (H0) that there is no trend and an alternative hypothesis (H1) that the trend is non-zero. This approach detects significant trends in both directions, regardless of the sign of the trend[84].

Similarly, the statistical significance of the differences between control (ctrl) runs and sensitivity experiments (sens) are calculated using the Welch's two-sample t-test[85]. At each grid point, the test statistic is calculated using

$$t = \frac{\bar{x}_{\text{ctrl}} - \bar{x}_{\text{sens}}}{\sqrt{\frac{s^2_{\text{ctrl}}}{n_{\text{ctrl}}} + \frac{s^2_{\text{sens}}}{n_{\text{sens}}}}} \tag{3}$$

with the Welch-Satterthwaite estimate of degrees of freedom defined as

$$\nu = \frac{\left(\frac{s^2_{\text{ctrl}}}{n_{\text{ctrl}}} + \frac{s^2_{\text{sens}}}{n_{\text{sens}}}\right)^2}{\frac{\left(\frac{s^2_{\text{ctrl}}}{n_{\text{ctrl}}}\right)^2}{n_{\text{ctrl}}-1} + \frac{\left(\frac{s^2_{\text{sens}}}{n_{\text{sens}}}\right)^2}{n_{\text{sens}}-1}} \tag{4}$$

Two-sided $p$-values are then computed as

$$p = 2\left[1 - F_{t,\nu}(|t|)\right], \tag{5}$$

where $\bar{x}$, $s$, and $n$ denote the sample mean, standard deviation, and sample size for each ensemble (e.g., control and sensitivity experiments for each model), and $F_{t,\nu}$ is the cumulative distribution function of the Student's $t$-distribution with $\nu$ degrees of freedom. Statistical significance is assessed at the 90% confidence level.

### Climate models and sensitivity experiments

The coupled models used for the sensitivity experiments are the Climate Forecast System version 2 (CFSv2 - CFS hereafter)[66], developed by the National Centers for Environmental Prediction (NCEP), and SINTEX-F2 (SINTEX hereafter)[65]. Both coupled models have good skills in simulating and seasonally forecasting the tropical climate variability. As an illustration, CFSv2 is the current operational climate prediction model for seasonal prediction in the US (at NCEP) and in India as part of the Monsoon Mission program (see http://www.tropmet.res.in/monsoon/), and SINTEX-F2 is also a standard tool for seasonal forecasting[86] and simulating tropical variability[11,65,87].

The atmospheric component of CFS is the Global Forecast System (GFS), which operates at T126 spectral resolution (approximately $0.9° \times 0.9°$) with 64 hybrid sigma-pressure levels. GFS is used in the NCEP2 reanalysis and the Twentieth Century Reanalysis (20CR), both developed at NOAA in the US. The ocean component (MOM model) has a horizontal resolution of $0.25°–0.5°$, 40 vertical levels, and includes a sea ice model. The atmosphere and ocean exchange heat and momentum fluxes every 30 minutes without any flux adjustment or correction. Further details on the CFS model and the configurations used here are provided in Saha et al.[66] and Terray et al.[11].

The atmospheric component of SINTEX is ECHAM5.4, which operates at T106 spectral resolution (approximately $1.125° \times 1.125°$) with 31 hybrid sigma-pressure levels. The ocean component (NEMO model) has a horizontal resolution of $0.5° \times 0.5°$, 31 unevenly spaced vertical levels, and includes a sea ice model. The atmosphere and ocean exchange heat and momentum fluxes every 2 hours without any flux adjustment or correction. Further details on the SINTEX model and the configurations used here are provided in Masson et al.[65] and Terray et al.[11].

In all experiments described below, CFS and SINTEX were run with fixed atmospheric $CO_2$ concentrations corresponding to present-day conditions. For each model, two sets of control simulations were performed. In the first set, the standard model configurations (ctrl) described in Saha et al.[66] and Masson et al.[65] were used, with integrations of 80 years for CFS and 210 years for SINTEX. A second set of control simulations employed an upgraded land albedo configuration (MODIS-ctrl) following Terray et al.[11], with integration lengths of 60 years for CFS and 110 years for SINTEX. In the standard atmospheric configurations of GFS and ECHAM5.4, the atmospheric components of CFS and SINTEX, respectively, snow-free background surface land albedo climatologies are prescribed as fixed boundary conditions. These climatologies are derived from isolated field measurements and outdated vegetation classifications[88,89]. This approach differs from that used in many Earth System Models and CMIP models, in which snow-free diffuse and direct surface albedos are typically parameterized based on vegetation and soil properties that evolve in time. Following Terray et al.[11], the MODIS-ctrl configurations replace the original snow-free background land albedo climatologies with satellite-derived estimates based on the Moderate Resolution Imaging Spectroradiometer (MODIS) gap-filled white-sky (diffuse) albedo product (MCD43GF-v5), averaged over the 2003-2013 period[90]. In addition, a seasonal cycle is introduced in the snow-free background land albedo climatology for ECHAM5.4, whereas the original climatology was time-invariant[89]. For more technical details on the original albedo parameterizations in the GFS and ECHAM5.4 and on the updated versions used here, readers are referred to[11,88,89]. Surface albedo exerts a strong control on the Earth's radiation budget, and climate models are therefore highly sensitive to its specification. Importantly, the updated MODIS-ctrl configurations substantially improve the simulation of tropical climate in both CFS and SINTEX[11]. The use of two coupled models and multiple configurations thus enables a robust assessment of the simulated responses.

These control simulations were compared with a set of parallel sensitivity experiments designed to assess the role of the land-sea thermal gradient in observed precipitation trends and to attribute the associated rainfall and circulation responses to specific land regions in the NH. In the first set of sensitivity experiments (zero-albedo runs), the snow-free background land albedo used in GFS and ECHAM5.4 was set to zero globally, and the models were integrated for 30 years (CFS) and 60 years (SINTEX). In a second set of experiments (desert-albedo runs), the updated MODIS background albedo climatologies were retained, but the snow-free background land albedo was additionally reduced by 20% over the Sahara-Arabian-Pakistan-Thar desert region (15°-40°N, 20°W-75°E), which was identified as a key source region for the tropical precipitation and circulation responses in the zero-albedo simulations. For these desert-albedo experiments, CFS and SINTEX were likewise integrated for 30 and 60 years, respectively. All the simulations performed with the two models are summarized in Table 1 and the distribution of surface albedo produced in the different simulations are illustrated in Supplementary Fig. 16.

For each model, differences between the sensitivity experiments and their corresponding control simulations (ctrl or MODIS-ctrl) are interpreted as the model response to changes in snow-free background land albedo and the associated modifications of the land-sea thermal gradient at global or regional scales. All analyses exclude the first 10 years of the SINTEX simulations and the first 5 years of the CFS simulations to allow for coupled model spin-up. Excluding a longer initial period (15 or 20 years) does not affect the results, consistent with the rapid atmospheric adjustment to land albedo perturbations, which occurs within a few months and stabilizes quickly in all sensitivity experiments.

Finally, CFS and SINTEX were integrated for different durations in the control and sensitivity experiments for two main reasons. First, the coupled system in CFS reaches equilibrium substantially faster than in SINTEX, allowing for shorter integrations after spin-up. Second, practical computational constraints limit the feasible integration length of CFS: its atmospheric component (GFS) is not efficiently parallelized, which substantially slows the coupled model and makes very long integrations computationally prohibitive.

## Data availability

The ERA5 data set used in this study is available to download from https://cds.climate.copernicus.eu/[76]. The CMIP6 data is available to download from https://digital.csic.es/handle/10261/332744[91]. The CFS model is available to download from https://cfs.ncep.noaa.gov/cfsv2/downloads.html and the SINTEX model is available at https://forge-web.ipsl.upmc.fr/sinext/browser/trunk. The processed climate model sensitivity experiment data and the data used to generate the figures are available at https://github.com/ligin1/Tropical-Precipitation-Response-to-Anthropogenic-Climate-Change-in-Recent-Decades[92].

## Code availability

Figures shown in this study are plotted using Python (https://www.python.org/). The Python code used for the analyses in this study is available at https://github.com/ligin1/Tropical-Precipitation-Response-to-Anthropogenic-Climate-Change-in-Recent-Decades[92].

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

## Acknowledgements

This work was supported by the Natural Environment Research Council [Grant NE/S007210/1]. Pascal Terray is funded by Institut de Recherche pour le Développement (IRD, France). K.P. Sooraj is funded by the CCCR and IITM, which are fully funded by the Ministry of Earth Sciences, Government of India. This project was provided with computing HPC and storage resources by GENCI at TGCC (France), thanks to the grants 2024-A0170113051 and 2025-A0190113051 on the supercomputer Joliot Curie's SKL and ROME partitions. This work was also supported by a French government grant managed by the 'Agence Nationale de la Recherche' under the 'investissements d'avenir' program (reference "ANR-21-ESRE-0051"). It was granted access to the MesoNET resources center and the MesoNET project under the allocation m24050. K.P. Sooraj sincerely thank the Director, IITM, for support during the research study. The authors gratefully acknowledge the financial support given by the Earth System Science Organization, Ministry of Earth Sciences, Government of India, to conduct parts of this research under the National Monsoon Mission (Grant number: MM/SERP/ CNRS/2013/INT-10/002, Contribution number: MM/PASCAL/RP/08). The authors gratefully acknowledge Lijo Abraham Joseph for his valuable feedback on improving figure quality and orientation, as well as for his help in correcting errors in the Python code used for analysis and figure production.

## Author contributions

L.J.: Conceptualization, Formal Analysis, Investigation, Methodology, Resources, Software, Supervision, Validation, Visualization, Writing - original draft; P.T.: Conceptualization, Formal Analysis, Investigation, Methodology, Resources, Software, Supervision, Writing - original draft, Writing - review and editing; K.S.: Resources, Writing - review and editing; S.M.: Resources, Writing - review and editing.

## Competing interests

The authors declare no competing interests.
