## [Transparent Peer Review file · Nature Communications]

Tropical Precipitation Response to Anthropogenic Climate Change in Recent Decades

Corresponding Author: Mr Ligin Joseph

Version 0:

Reviewer comments:

Reviewer #1

(Remarks to the Author)

The authors conduct a comprehensive diagnosis and attribution analysis of tropical precipitation changes using observational datasets, reanalysis products, and CMIP6 model simulations. The study covers spatial trends, dynamical–thermodynamical decomposition, and temporal trend assessments. The observational analysis is particularly thorough and convincing, providing a valuable contribution to understanding recent decades of tropical precipitation changes. The paper also offers a clear critique of the limitations of the CMIP6 multi-model ensemble in reproducing the observed trends.

However, before the paper can be published, I have the following suggestions, particularly regarding the sensitivity experiments

1. When discussing the dominance of dynamical contributions over the simple C–C scaling, the key role of land–sea thermal contrast, the inadequacy of the conventional “warmer–wetter” framework to explain recent observed patterns, and the enhanced sea-to-land moisture transport, it would strengthen the discussion to cite Lan et al. (2024), which presents similar arguments.

Lan, C. W., Chen, C. A., & Lo, M. H. (2024). The role of atmospheric stabilities and moisture convergence in the enhanced dry season precipitation over land from 1979 to 2021. *Journal of Climate*, 37(9), 2881–2893.

2. The manuscript does not explain why the two models, CFS and SINTEX, were run for different integration lengths in the control simulations and in the sensitivity experiments. It is recommended to provide a brief explanation of the rationale behind this design choice.

3. This paper mentioned the CMIP6 MME for failing to reproduce the observed tropical precipitation trends during the satellite era (1979–2024), however, it does not present direct comparisons of the CFS and SINTEX control simulations with the observations or reanalysis, such as bias maps. Including such evaluations would highlight the representativeness and reliability of the two selected models used in the sensitivity experiments.

4. The observational analyses identify the land–sea thermal contrast, Indo-Pacific Warm Pool (IPWP) SST warming, and the inter-hemispheric thermal gradient as major factors associated with the observed tropical precipitation trends. Among these, land–sea thermal contrast and IPWP SST warming show the strongest regression signals and best reproduce the observed precipitation and circulation patterns.

However, the sensitivity experiments in the paper focus only on a single factor—an idealized setup that maximizes land warming relative to ocean warming. This inevitably leads to conclusions that differ from the traditional view emphasizing oceanic forcing and also produces precipitation changes far larger than those observed.

It is recommended to provide additional information, such as:

- > performing further sensitivity experiments with progressively varied albedo forcing or other external drivers to explore the gradual response of precipitation and circulation to different degrees of thermal contrast;
- > analyzing energy-budget and albedo diagnostics, linking surface–atmosphere energy balance, large-scale circulation, and the observed trends, to better verify the consistency between the idealized experiments and real-world evidence.

Reviewer #2

(Remarks to the Author)

Review of “Tropical precipitation response to anthropogenic climate change in recent decades” by Joseph et al. for publication in *Nature Communications*.

This study addresses a timely and important issue: how tropical precipitation has responded to global warming during the satellite era (1979-2024). The authors use reanalysis products, observations, CMIP6 simulations, and targeted sensitivity experiments to argue that increased land-sea temperature contrasts (and Indo-Pacific warm pool intensification), rather than mechanisms such as “wet get wetter” (WeGW), warm get wetter (WaGW), or the “direct effect of CO₂ forcing” (DeCO₂), best explain the observed trends in tropical precipitation and circulation. The main result of the paper, that increased land-sea temperature contrast can drive large-scale precipitation patterns in the tropics rather than acting as a passive forcing, is noteworthy.

Despite the potential importance and novelty, there are major issues that need to be resolved adequately. Here are my detailed evaluations (not in any particular order):

1. Much of the paper (before Sec 2.4) is devoted to correlating observed changes in precipitation and circulation with other parameters, such as different temperature-based indices. The actual mechanisms were primarily addressed through sensitivity experiments (Sec. 2.4), where the land surface albedo was modified to zero, thereby increasing land temperatures and enhancing the land-sea temperature gradient. The adopted procedure, however, suffers from multiple drawbacks:

(i) Setting land albedo to zero is an extreme case that would likely trigger unknown non-linear processes and interactions that may not be seen or relevant to the real world. I would have expected to see more moderate perturbations by gradually changing the albedo by 10%, 20%, ..., ... compared to the control simulation. The results from such experiments will also indicate if the response is linear or non-linear.

As it stands, one may argue that these simulations were useful to test sensitivity but not attribution. Note that the change in land temperature under zero albedo is more than an order of magnitude larger than the observed change in land temperature.

(ii) It seems the results were based on a single realization of each experiment (control and zero albedo). The lack of a reasonable number of ensemble members is a serious limitation that reduces confidence in the robustness and range of simulated responses.

(iii) It is unclear whether land surface emissivity was modified when albedo was set to zero. Because reflectivity + absorptivity (emissivity) + transmissivity should be 1. Assuming negligible transmissivity of the surface, was the land surface treated as a perfect black body (i.e., perfect emitter)? This is important information that is missing, but critical to understand the model simulations.

(iv) Given the non-linear nature of the Earth system, it is not clear why sensitivity experiments were not considered related to other factors that also showed connections with the observed changes in precipitation and circulation, such as hemispheric temperature difference and Indo-Pacific warm pool temperature. (The authors specifically mention on page 13, “Overall, our results highlight three correlated, climate-change–related indices, the inter-hemispheric and land–sea thermal gradients and the intensification of the Indo-Pacific warm pool, as leading candidates driving the observed tropical precipitation trends”. However, right after that, they mention conducting albedo perturbation experiments to test their hypothesized role of land-sea temperature contrasts.

(v) I was wondering whether the zero albedo experiment truly separated the effect of land-sea contrast from the interhemispheric temperature difference. Given that there are more lands in the NH, the zero albedo experiment will lead to increased land temperature in both hemispheres, and hence warmer NH than SH due to more land in the NH.

(vi) Recent observed increase in land temperature is arguably due to a multitude of processes, and not due to an increase in albedo alone. So, it was not clear why albedo reduction was chosen to perform the sensitivity experiments instead of considering other possibilities, in particular, those that have been found to enhance observed land surface temperature (such as adding surface heat flux, changing atmospheric longwave emissivity).

(vii) How quickly did SST and land-temperature patterns emerge after albedo perturbation? This may help in diagnosing whether land warming drives ocean changes (or vice versa?).

2. The rationale behind using a simple two-layer moisture budget analysis instead of a vertically integrated moisture budget analysis was not clear. What would be the typical error for considering this simplified approach? Given that this paper is focused on precipitation, I was not expecting a simplified approach. (The use of this simplified approach in past studies cannot be the sole reason behind choosing it.)

Is there a possibility that this two-layer approach using near-surface humidity and mid-tropospheric vertical winds biases towards dynamics than thermodynamics?

Also, it seems ERA5 doesn't fully close the moisture budget. Therefore, can it impact your results and conclusions?

3. About climate indices (Sec 4.4): Multiple temperature-based indices were used. I was wondering if there are any reasons to not include PDO related SST change. This also brings the question whether the analysis period (1979-2024) is sufficient to distinguish between forced signals and decadal variability, given that PDO influence could be substantial.

4. The second half of Sec 2.4 and Section 3 (Summary and Discussion) are repetitive, and should be streamlined. But, more

importantly, I felt that much of the discussion and its interpretation of the sensitivity experiments were qualitative.

5. Your regression involved rainfall and velocity potential with trends in several parameters, such as GMP, land-sea contrast, interhemispheric temperature gradient, etc. Given that many of these parameters themselves are highly correlated, it seems that one will need multivariate regression to infer the dominance of one index over another.

6. I felt the discussion on Walker circulation strengthening/weakening was somewhat qualitative, and based on upper-level velocity potential and surface winds. Is there an index or quantitative approach to estimate the strength of Walker circulation?

7. Overall presentations need significant improvements throughout the paper, such as:

(i) Subsections in “Results” are named as “Trend Analysis” (sec 2.1), “Time Series and Regression Analysis” (sec 2.3) – I would avoid naming sections following statistical methods. Instead, sections should be named after the scientific content or processes discussed in that section.

(ii) It was very difficult to go through several figures in the manuscript as well as in the supporting document because of various reasons, like very small fonts, small panel sizes, etc. Figures need to be thoroughly modified for improving clarity.

(iii) Please add units to the figure captions. Check all.

8. There were discussions about dynamic and thermodynamic controls on changes in precipitation – this is prevalent in the literature as well. I was curious to know to what extent dynamic and thermodynamic effects are independent of each other!! Is it possible to separate their influences in sensitivity experiments?

9. Apart from the surface forcing (land-sea temperature contrast), I would imagine that the atmospheric stability over land and ocean would also play a role, but I did not seem to find anything on stability – any thoughts?

10. Observed precipitation trend (1979-2024) in the tropics (Fig. 1a) seems to be statistically insignificant. What does it mean in the context of the sensitivity experiments that show the simulated precipitation trend is largely statistically significant (Fig. 5)?

Other comments:

Introduction: 1st line: State of Climate to State of the Climate

2nd line: mean surface air temperature to mean near-surface air temperature

3rd paragraph, first sentence: “Taken together” The sentence is not clear, rewording needed.

Paragraph starting with “In this context...” and the next paragraph: very long, consider splitting. Check all.

Page 3, line 6: theories to hypotheses

Page 3, 2nd para, line 11: remove “actually”

Page 4, line 3: sensitivity coupled experiments to coupled sensitivity experiments

Page 4, sec 2.1, line 8: SST trends or SST patterns

Page 5: (d) 850-hPa wind trend (e) 200-hPa velocity potential trend
Wind vectors are not legible at all.

Page 6, line 7: It seems Supp Fig. 4 is introduced before Supp fig. 3. Check.

Page 6, Line 13: “Yet, importantly,” Not clear, rewording needed.

Page 6, line 11 from bottom: also arises to may arise

Page 6, line 14 from bottom: The sentence, “As this contrast” is very long. Consider splitting.

Page 6, line 11 from below: what is “effective radiative forcing” here?

Page 7, lines 2-3: However, the Southern Oceanremove because you are talking about it just a few sentences later.

Page 8, line 5: In summary to Overall

Page 8, line 12 from bottom: "However, the reconstructed trend" – Was not clear which panels are being compared

Page 11, 2nd para, "Its regression" – not clear which regression

Page 13, line 8 from bottom: "existence of feedbacks..." – I was curious what these feedbacks may be.

Page 16, sec 3, 2nd para: " These results suggest three possible scenarios:" Can they all be true?

Page 21, "by fitting a polynomial of degree 1 or 2" – any reason not to consider polynomials of higher degrees? Also, I see a window width of 6 years. Does it remove/smooth ENSO or decadal variability?

Supplementary Fig. 4: Did you use the ONI index to separate EL Niño and La Niña phases? Oftentimes, the phases change within the same year, so considering a full year as an El Niño or La Niña year may not be well-suited here.

Also, mention how the inter-hemispheric thermal gradient was calculated (NH minus SH or SH minus NH).

Reviewer #3

(Remarks to the Author)

Formal review of "Tropical Precipitation Response to Anthropogenic Climate Change in Recent Decades" by Ligin Joseph et al.

Article reference: NCOMMS-25-70198

The authors study a range of climate responses using ERA5 reanalysis, CMIP6 simulations, targeted sensitivity experiments by GCMs with a focus on tropical precipitation responses. They motivate their study by current climate models which show large precipitation biases in the tropics. They also describe several current "theories", among these "wet-get-wetter" (WeGW) and "warm-get-wetter" paradigms. Within WeGW and weakening of atmospheric circulations, e.g., Walker and Hadley cells, due to increased atmospheric moisture and reduced radiative cooling is postulated. They point out that there are discrepancies between model and observational findings, where the latter show a strengthening of the Walker circulation despite significant surface warming. Also, CMIP models show a decrease in the equatorial SST gradient during recent decades, whereas observations show an increase.

Citing Shrestha, S., Soden, B.J.: Anthropogenic weakening of the atmospheric circulation during the satellite era. *Geophysical Research Letters* 50(22) (2023) (Ref 39) the authors state that global mean circulation might indeed be weakening due to reduction in the Walker circulation.

Joseph et al. further caution that precipitation response to global warming is a complex phenomenon, with an ongoing debate on whether circulation changes might be a main cause. They acknowledge that, how the water cycle might have changed in the course of recent decades, remains poorly known.

In their results they distinguish three main sections:

1. A discussion on spatial trends in key variables: precipitation, 2m air temperature, SST, 850 hPa winds, as well as 200hPa velocity;
2. They then decompose precipitation changes into dynamic and thermodynamic components;
3. They finally discuss eight surface temperature indices in observations and simulations.

In detail, within their trend analysis they find that ERA reanalysis data roughly agrees with other observational and reanalysis datasets regarding overall trends. CMIP6 model simulated precipitation shows very different trends from the ERA data, e.g., with weak temperature changes in the tropics but strong precipitation changes.

Pasted figure 1: ERA-derived precipitation changes

Pasted figure 2: CFS (left column) and SINTEX (right column)

Regarding the attribution to dynamics vs. thermodynamics, in the following subsection they demonstrate that the observed tropical rainfall trends mainly appear to be controlled by dynamical processes rather than thermodynamics. They claim that surface temperature gradients play a key role in driving the atmospheric circulation. They finally discuss several indices, even though the necessity for this discussion did not become obvious to me, it seems not needed here.

More interestingly, they attempt to attribute the precipitation changes seen in ERA data to other variables, in particular near-surface temperature. They isolate trends in land-sea warming contrast as a potential cause of circulation changes and thus precipitation pattern shifts. They further use two global coupled models, namely CFS and SINTEX to re-enact (and likely exaggerate) the effect of land-sea temperature changes by reducing the over-land albedo to

zero. Indeed, in some regions the patterns of precipitation changes produced by their modeling results are not too different from the ERA-derived results (see the comparison of the panels in Fig. 1a vs. Materials Fig. 2c,d). This is especially true for precipitation over the Pacific Ocean where some notable similarities exist. Yet, in most other regions of the globe the changes in precipitation seem much less similar and are often even of opposite sign, e.g. most continental regions and the Atlantic Ocean.

In summary, I agree that CMIP models do not show a strong similarity with ERA-derived historical precipitation trends. I appreciate that the authors attempted an idealized sensitivity study using their own model simulations to emphasize certain aspects of precipitation change patterns. I am currently not convinced that the changes they simulate sufficiently describe the ERA-derived changes in precipitation, even though some similarity exists for the Pacific Ocean - but most other regions disagree.

Version 1:

Reviewer comments:

Reviewer #1

(Remarks to the Author)

The authors have addressed my questions and incorporated the corresponding revisions into the manuscript. I have no further questions.

Reviewer #2

(Remarks to the Author)

2nd round review of "Tropical precipitation response to anthropogenic climate change in recent decade" by Joseph et al. for publication in Nature Communications.

The authors have adequately addressed my main concerns. Here are a few suggestions that could further improve the clarity of the paper.

1. (ii) I believe that further experiments have provided more confidence on the results. In this regard, the authors argue, "We believe that this strategy is the right one in the present context as we deal only with mean-state change, not variability." – This statement seems to hold only if variability doesn't influence the evolving mean-state. However, in a changing climate, the mean state and variability may evolve together, and variability could potentially feed back onto mean state changes. So, the question is, how do you know that changes in variability do not affect the evolving mean state in this context?

1. (iv) Part of your response to comment (iv) would be helpful to include in the paper for improved clarity.

1. (v) Same here. A sentence or two related to why isolation was not the goal should be included in the paper.

1. (vii) The authors state: "The land temperature and SST patterns emerge very quickly, in a few months, in both the idealized zero-albedo experiments and the new idealized desert-albedo experiments now included in the revised manuscript." – What I was looking for was if the land temperature patterns emerged earlier than SST patterns. If land and SST patterns emerge simultaneously, how can land surface warming be interpreted as the driver of SST changes?

Reviewer #3

(Remarks to the Author)

Version 2:

Reviewer comments:

Reviewer #2

(Remarks to the Author)

The authors have adequately addressed my comments. Congratulations!

Reply to Reviewer 1:

The authors conduct a comprehensive diagnosis and attribution analysis of tropical precipitation changes using observational datasets, reanalysis products, and CMIP6 model simulations. The study covers spatial trends, dynamical–thermodynamical decomposition, and temporal trend assessments. The observational analysis is particularly thorough and convincing, providing a valuable contribution to understanding recent decades of tropical precipitation changes. The paper also offers a clear critique of the limitations of the CMIP6 multi-model ensemble in reproducing the observed trends.

However, before the paper can be published, I have the following suggestions, particularly regarding the sensitivity experiments:

Answer: Thank you very much for your constructive and valuable comments, which have helped us to significantly improve our revised manuscript. Below we provide detailed responses to the reviewer’s suggestions, with page and paragraph numbers referring to the revised manuscript. All changes are highlighted in the tracked-changes version. All references cited in this response are provided in the main text of the manuscript.

- 1. When discussing the dominance of dynamical contributions over the simple C–C scaling, the key role of land–sea thermal contrast, the inadequacy of the conventional “warmer-wetter” framework to explain recent observed patterns, and the enhanced sea-to-land moisture transport, it would strengthen the discussion to cite Lan et al. (2024), which presents similar arguments.*

Answer: We thank the reviewer for this suggestion. We have now cited Lan et al. (2024) wherever applicable. In addition, we have also added new references relevant to our study in the revised manuscript to better put our work in the context of related literature as requested by the Editor. This includes :

- Obarein and Lee (2025), who show that ERA5 reproduces key features of recent global precipitation trends, which further justifies the use of ERA5 in our investigations;
- Aumann et al. (2024), who found a northward shift and narrowing of the ITCZ during recent decades from atmospheric infrared sounder data, which are fully consistent with our analysis of ERA5 rainfall

- 2. The manuscript does not explain why the two models, CFS and SINTEX, were run for different integration lengths in the control simulations and in the sensitivity experiments. It is recommended to provide a brief explanation of the rationale behind this design choice.*

Answer: Thank you for this comment. These details are now provided in the revised manuscript (Section 4.7, last paragraph). Note also that new simulations are now used in the revised manuscript, as summarized below.

The CFS and SINTEX control simulations in their standard configurations (mentioned as ctrl in the revised manuscript) are long runs that serve as a reference for the other sensitivity experiments presented in the revised manuscript. The CFS ctrl simulation is shorter than the

SINTEX ctrl mainly for two reasons. The first is that the spin-up of this model is much faster than in the SINTEX model. The second one is practical; the CFS coupled model is, in fact, very slow to run since its atmospheric component (e.g., the GFS atmospheric model) is not efficiently parallelized. This slows down the whole coupled model, making it very challenging to perform long coupled simulations with the CFS.

However, to further demonstrate the robustness of our results, we now also use two new CFS and SINTEX control simulations using upgraded versions of the two models (mentioned as MODIS-ctrl in the revised manuscript and described in Terray et al. 2018), which can also be compared with our idealized zero-albedo sensitivity experiments.

The lengths of the CFS and SINTEX sensitivity experiments (e.g., the idealized zero-albedo experiments in our original submission and both the idealized zero-albedo and desert-albedo experiments in our revised manuscript, see below for more details) also take into account these constraints and were determined to obtain a stable response to the imposed albedo perturbations in the different runs. As an illustration, the response to the albedo perturbations in the CFS sensitivity experiments emerges only in a few months and is completely stable after only 10 years of simulation, thanks to the large amplitude of the response in the CFS model, but 30 years of simulation were still finally computed as a compromise between robustness and efficiency (Terray et al. 2018, Sooraj et al. 2019). Similarly, the response to the albedo perturbations in the SINTEX sensitivity experiments is completely stable after 30 years of simulation, but the simulations were extended up to 60 years, again as a compromise between robustness and efficiency. These details are now provided in Section 4.7 “Climate Models and Sensitivity Experiments” of the revised manuscript.

Finally, note that in addition to the new simulations now used in the revised manuscript, we have also updated the SINTEX idealized zero-albedo simulation, which is now fully consistent with the one performed with the CFS model in terms of the amplitude of the response. In the previous SINTEX idealized zero-albedo experiment, the snow-free land background albedo was not set exactly to zero because of a coding error; this is now corrected.

All the simulations used are now summarized in Table 1 of the revised manuscript, and the Climate Models and Sensitivity Experiments Subsection 4.7 has been fully revised to reflect these modifications. Importantly, due to size constraints, we show in the main manuscript only the SINTEX results, and all the CFS results are now moved to the Supplementary document.

3. This paper mentioned the CMIP6 MME for failing to reproduce the observed tropical precipitation trends during the satellite era (1979–2024), however, it does not present direct comparisons of the CFS and SINTEX control simulations with the observations or reanalysis, such as bias maps. Including such evaluations would highlight the representativeness and reliability of the two selected models used in the sensitivity experiments.

Answer: Thank you for this suggestion. We have now included a section and a new figure showing the model's bias for the different control simulations in the Supplementary document (SI Section 2 and Fig: 9), and a short paragraph focusing on the biases of both models is now included in the revised main manuscript (in subsection 2.1). Note also that these two models have been heavily used in the past literature and performed rather well for the coupled

simulation of tropical variability. For example, the CFS model is the model used in all the past NCEP reanalyses and for seasonal forecasting in the US and India. Similarly, the SINTEX couple model is also used for seasonal forecasting in Japan (e.g., JAMSTEC). This is also now summarized in the revised manuscript.

As expected, and shown in the Supplementary Document, the SINTEX and CFS models share many biases similar to the CMIP models, which explain why we used albedo perturbations of strong amplitude (e.g., the idealized zero-albedo experiments), first to illustrate the potential role of the amplified land-sea contrast.

4. The observational analyses identify the land–sea thermal contrast, Indo-Pacific Warm Pool (IPWP) SST warming, and the inter-hemispheric thermal gradient as major factors associated with the observed tropical precipitation trends. Among these, land–sea thermal contrast and IPWP SST warming show the strongest regression signals and best reproduce the observed precipitation and circulation patterns.

However, the sensitivity experiments in the paper focus only on a single factor—an idealized setup that maximizes land warming relative to ocean warming. This inevitably leads to conclusions that differ from the traditional view emphasizing oceanic forcing and also produces precipitation changes far larger than those observed.

It is recommended to provide additional information, such as:

> performing further sensitivity experiments with progressively varied albedo forcing or other external drivers to explore the gradual response of precipitation and circulation to different degrees of thermal contrast;

> analyzing energy-budget and albedo diagnostics, linking surface–atmosphere energy balance, large-scale circulation, and the observed trends, to better verify the consistency between the idealized experiments and real-world evidence.

Answer: Thank you for these constructive recommendations.

Energy-budget and albedo diagnostics are now used in the revised Subsection 2.4 (focusing on the sensitivity experiments) to better verify the consistency between the idealized experiments and real-world evidence, as requested.

In response, we have also conducted additional sensitivity experiments with both models in which we reduce surface albedo by 20% only over the Sahara, Arabia, and Middle East arid regions (these new simulations are referred to as the desert-albedo experiments in the revised manuscript). We choose to perturb albedo only in these arid regions of the Northern Hemisphere because of the following reasons:

- First, these regions make a major contribution to the net radiative flux at the top of the atmosphere—specifically to the balance between net downward shortwave radiation and outgoing longwave radiation—and therefore strongly influence the tropical land–sea thermal contrast in the idealized zero-albedo experiments. This is now explicitly demonstrated in the revised Section 2.4.
- Second, desert amplification has been identified as a key feature of recent observed climate change (Wei et al. 2017; Sooraj et al. 2021). As a result, the surface warming and the increased surface sensible heat flux associated with the observed desert

amplification represent a physically relevant target for testing the gradual strengthening of the land–sea thermal gradient and the observed tropical climate response. Obviously, the mechanisms behind the surface warming over these arid regions in the idealized desert-albedo experiments are different from those operating in the real climate (Wei et al. 2017; Sooraj et al. 2021), but we believe that these new experiments are still useful to assess the dynamical climate response to this surface warming and the associated increase of the surface sensible heat flux as observed in the real climate.

Interestingly, the results show precipitation and low- and upper-level circulation responses that are qualitatively similar to those obtained in the idealized zero-albedo experiments, but with substantially weaker magnitudes as expected (but still statistically significant) and much more in line with the observed rainfall and circulation trends (e.g. compare Figs. 1 and 5; SI Figs. 10, 11, 13). This confirms that the land–sea thermal contrast plays a leading role in shaping the observed tropical precipitation trends and further highlights the importance of warming over the major desert regions of the Northern Hemisphere in driving the associated rainfall and circulation responses. The key-role of the warming over the major desert regions of the Northern Hemisphere in the idealized experiments is consistent with the fact that these areas maximize the top-of-atmosphere net radiative differences between the idealized experiments and the control runs for both CFS and SINTEX (see Fig. 6 in the revised manuscript and SI Figs. 12, 14, and 15).

Regarding the interhemispheric thermal gradient, as noted in Subsection 2.3, it is highly correlated with the land–sea thermal gradient in both ERA5 and CMIP datasets (correlation coefficient ≈ 0.99). Because of this near-collinearity and the key role of the subtropical deserts of the Northern Hemisphere highlighted by the new idealized desert-albedo experiments, we still focused our analysis on the land–sea gradient and skipped the inter-hemispheric gradient from further analysis.

Note, however, we still focus on a single factor, e.g., the land-sea contrast and the desert amplification, in the revised manuscript, because this factor was almost unexplored in past studies, and also because testing the role of the Indo-Pacific warming pool requires a completely different coupled model setup (e.g., nudging SST coupled model experiments), which is left for future studies. Of course, this does not imply that the land-sea contrast is the only driver of the observed changes, and this is clearly highlighted now in the conclusion section of the revised manuscript. However, we hope that the Reviewer will be satisfied with these new idealized desert-albedo experiments, which provide a better consistency with the real-world evidence than the original idealized zero-albedo experiments and point to possible physical mechanisms behind the observed and simulated tropical climate response.

Reply to Reviewer 2:

This study addresses a timely and important issue: how tropical precipitation has responded to global warming during the satellite era (1979-2024). The authors use reanalysis products, observations, CMIP6 simulations, and targeted sensitivity experiments to argue that increased land-sea temperature contrasts (and Indo-Pacific warm pool intensification), rather than mechanisms such as “wet get wetter” (WeGW), warm get wetter (WaGW), or the “direct effect of CO₂ forcing” (DeCO₂), best explain the observed trends in tropical precipitation and circulation. The main result of the paper, that increased land-sea temperature contrast can drive large-scale precipitation patterns in the tropics rather than acting as a passive forcing, is noteworthy.

Despite the potential importance and novelty, there are major issues that need to be resolved adequately. Here are my detailed evaluations (not in any particular order):

Answer: Thank you very much for your constructive and valuable comments, which have significantly improved our manuscript. Below we provide detailed responses to the reviewer’s suggestions, with page and paragraph numbers referring to the revised manuscript. All references cited in this response are provided in the main text of the manuscript, if not, then at the end of this document.

- 1. Much of the paper (before Sec 2.4) is devoted to correlating observed changes in precipitation and circulation with other parameters, such as different temperature-based indices. The actual mechanisms were primarily addressed through sensitivity experiments (Sec. 2.4), where the land surface albedo was modified to zero, thereby increasing land temperatures and enhancing the land-sea temperature gradient. The adopted procedure, however, suffers from multiple drawbacks:*

(i) Setting land albedo to zero is an extreme case that would likely trigger unknown non-linear processes and interactions that may not be seen or relevant to the real world. I would have expected to see more moderate perturbations by gradually changing the albedo by 10%, 20%, ..., ... compared to the control simulation. The results from such experiments will also indicate if the response is linear or non-linear.

As it stands, one may argue that these simulations were useful to test sensitivity but not attribution. Note that the change in land temperature under zero albedo is more than an order of magnitude larger than the observed change in land temperature.

(ii) It seems the results were based on a single realization of each experiment (control and zero albedo). The lack of a reasonable number of ensemble members is a serious limitation that reduces confidence in the robustness and range of simulated responses.

(iii) It is unclear whether land surface emissivity was modified when albedo was set to zero. Because reflectivity + absorptivity (emissivity) + transmissivity should be 1. Assuming negligible transmissivity of the surface, was the land surface treated as a perfect black body

(i.e., perfect emitter)? This is important information that is missing, but critical to understand the model simulations.

(iv) Given the non-linear nature of the Earth system, it is not clear why sensitivity experiments were not considered related to other factors that also showed connections with the observed changes in precipitation and circulation, such as hemispheric temperature difference and Indo-Pacific warm pool temperature. (The authors specifically mention on page 13, “Overall, our results highlight three correlated, climate-change–related indices, the inter-hemispheric and land–sea thermal gradients and the intensification of the Indo-Pacific warm pool, as leading candidates driving the observed tropical precipitation trends”.

However, right after that, they mention conducting albedo perturbation experiments to test their hypothesized role of land-sea temperature contrasts.

(v) I was wondering whether the zero albedo experiment truly separated the effect of land-sea contrast from the interhemispheric temperature difference. Given that there are more lands in the NH, the zero albedo experiment will lead to increased land temperature in both hemispheres, and hence warmer NH than SH due to more land in the NH.

(vi) Recent observed increase in land temperature is arguably due to a multitude of processes, and not due to an increase in albedo alone. So, it was not clear why albedo reduction was chosen to perform the sensitivity experiments instead of considering other possibilities, in particular, those that have been found to enhance observed land surface temperature (such as adding surface heat flux, changing atmospheric longwave emissivity).

(vii) How quickly did SST and land-temperature patterns emerge after albedo perturbation? This may help in diagnosing whether land warming drives ocean changes (or vice versa?).

Answers: We thank the reviewer for their valuable and constructive comments. Below, we provide point-by-point answers to the concerns raised by the reviewer.

I. We agree with the reviewer that setting the snow-free background land albedo in the models to zero is an extreme case, and as such, the idealized zero-albedo experiments cannot be used for attribution.

In order to address both of these issues, e.g., the extreme temperature response in the idealized zero-albedo experiments and the attribution problem, we now show another sensitivity experiment with both models where we decreased the snow-free background land albedo by 20% over the Sahara, Arabia and Middle-East deserts only (these new experiments are referred as desert-albedo experiments in the revised manuscript, see Table 1 of the revised manuscript). We chose to perturb the background land albedo only in these regions mainly for two reasons:

- First, because they make a major contribution to the tropical land–sea thermal contrast and the associated atmospheric and radiation responses in the idealized zero-albedo experiments (see Figs. 5 and 6 in the revised manuscript and SI Figs.10-15 in the revised Supplementary Document). Note also that surface and Top-Of-Atmosphere (TOA) radiation budget diagnostics for the different experiments are now provided and discussed in Section 2.4 for a better illustration of physical mechanisms in action.

- Second, because desert amplification has been identified as a key feature of recent observed climate change (Wei et al. 2017; Sooraj et al. 2021).

In other words, these regions also provide a good candidate for attribution of the simulated changes and represent a physically relevant target for testing the gradual strengthening of the land–sea thermal gradient at the same time as requested by the reviewer.

Moreover, the results show precipitation and circulation responses that are qualitatively similar to those obtained in the idealized zero-albedo experiments, but with substantially weaker magnitudes as expected and now more inline with observations (e.g., compare Figs. 1, 5, SI Figs. 10, 11 with SI Fig. 13). This confirms that the land–sea thermal contrast in the tropical Northern Hemisphere plays a leading role in shaping the observed tropical precipitation trends and highlights the importance of warming over the major desert regions of the Northern Hemisphere in driving the rainfall and circulation responses, despite these arid regions are often considered as a passive element in the climate system (Sooraj et al. 2021). This is partly in agreement with some studies, which suggest that increased CO₂ over land dominates the regional response to climate change via energy input over land, e.g., over deserts where there is no cloud and water vapor masking (Shaw and Voigt 2016). Obviously, the mechanisms behind the surface warming over these arid regions in these new idealized desert-albedo experiments are different from those operating in the real climate (Shaw and Voigt 2016; Wei et al. 2017), but it still suggests that the dynamical and rainfall responses to the associated increased land-sea contrast can be similar.

- II. First, past experiences with both the CFS and SINTEX models indicate that the control simulations used here are long enough to provide a robust reference for the tropical mean states in these models (here recall that we deal only with mean-state changes over long periods). Furthermore, as we are dealing here with the climate response to (i.e., strong and fast responses which quickly reach a stable state) radiative perturbations, we don't think that the best way of demonstrating the robustness of the response is to use a single model with different members.

Instead, we choose to perform similar experiments with two completely different coupled models, which, in addition, parameterize the land surface albedo by completely different methods (Hou et al. 2002; Roeckner et al. 2003; Terray et al. 2018) to test the robustness of the results. We believe that this strategy is the right one in the present context as we deal only with mean-state change, not variability.

Last but not least, we have now added (i) another set of sensitivity experiments (e.g. the above idealized desert-albedo experiments), which confirms again the robustness of the results as the two models give again similar results and (ii) new control simulations with upgraded versions of the two models using improved albedo schemes, which are also now compared to the different sensitivity experiments (see Table 1 for an overview of all simulations now used in the revised manuscript). All these new simulations are in line with the original results. This further indicates that the results are robust.

- III. First, we note that, in ECHAM5.4 (the atmospheric component of SINTEX), the longwave emissivity assumes a constant value of 0.996 for all surfaces and spectral intervals, and that the longwave radiation flux for all surfaces is computed simply as a black body emission weighted by this emissivity. On the other hand, in the absence of

snow, the land surface albedo in ECHAM5.4 is simply equal to the snow-free background land surface albedo, which is fixed in time, but does vary in space (see Roeckner et al. (2003) for more details). In other words, in the standard configuration of ECHAM5.4, the surface albedo is not dependent on the emissivity. Consequently, we do not change the emissivity in all our SINTEX experiments as well, and the net surface radiation budget (R_{net}) over land is, in all cases, computed as:

$$R_{net} = (1 - \alpha)R_{sd} + e R_{ld} + e\sigma T_s^4$$

where R_{net} is the net surface radiation, α is the surface albedo, R_{sd} is the downwelling solar radiation, R_{ld} is the downwelling longwave radiation, e the emissivity, σ the Stefan-Boltzmann constant and T_s is the temperature of the surface layer. Furthermore, in the absence of snow, it is nothing but the snow-free background land albedo, which was set to zero in the zero-albedo experiments. In the standard configuration of ECHAM, this snow-free background albedo is a fixed background white-sky (e.g., diffuse) shortwave albedo over snow-free land surfaces (Roeckner et al. 2003). We now also show the surface albedo climatologies from the different simulations in SI Fig. 16 to illustrate these details.

In GFS (the atmospheric component of CFS), the approach used is similar (see Hou et al. (2002), for details), excepted that the emissivity do change according to a vegetation and soil classification, but this vegetation-soil classification is fixed in time, while the snow-free background land albedo do vary according to the season and is set independently from this vegetation classification. More precisely, the albedo parameterization in the GFS prescribes different snow-free background diffuse albedo for the ultraviolet and visible band (VIS, $<0.7 \mu\text{m}$) and the near-infrared band (NIR, $>0.7 \mu\text{m}$), and also their seasonal and spatial variations are taken into account in the form of fixed global maps for each season and each spectral band (Hou et al. 2002). Then, the direct beam albedo, which strongly depends on the Solar Zenith Angle (and thus on the time of the day and the geographical coordinates), is then parameterized from the prescribed diffuse albedo for each spectral band (e.g., VIS and NIR) using a scheme described by Hou et al. (2002). Although vegetation is not explicitly represented in the GFS albedo scheme, it can be regarded as implicitly resolved by these prescribed diffuse surface albedos, which vary in space and time, but this has no direct relationship with the vegetation-soil classification used by the emissivity. Consequently, and to be consistent with SINTEX, we also do not change the emissivity in the CFS simulations. This implies that the land surface is not treated as a pure perfect black body in the idealized zero-albedo experiments with both models.

Finally, note that both ECHAM and GFS separately compute at each time step albedos for soil-vegetation and snow, and then estimate the total albedo of a grid box as an average of these albedos weighted by the representative area fractions. This explains why the land surface albedo is not zero at high latitudes in the zero-albedo experiments illustrated in SI Fig. 16.

In summary, the diffuse background albedo over snow-free land surfaces in both GFS and ECHAM5.4 may be considered as a fixed boundary land condition and stays constant for each grid-point (and each season), and this is these background snow-free albedos (2 for GFS and 1 for ECHAM5.4), which are set to zero in our idealized zero-albedo experiments with the two coupled models or updated with MODIS data in the MODIS-ctrl simulations. The same applies to the idealized desert-albedo experiments now included in the revised

manuscript and discussed above. See Table 1 and Section 4.7 of the revised manuscript for a summary of these different experiments.

- IV. As noted in Section 2.3 of the manuscript, the inter-hemispheric thermal gradient is almost perfectly correlated with the land–sea thermal gradient in both ERA5 and CMIP6 (correlation ≈ 0.99). This strong collinearity is expected because the Northern Hemisphere contains substantially more land area than the Southern Hemisphere. As a result, any perturbation that amplifies the land–sea contrast inherently alters the inter-hemispheric temperature difference as well, and vice versa. Because of this near-degeneracy in the climate system, it is difficult to fully disentangle the influences of the land–sea gradient from the inter-hemispheric gradient using independent sensitivity experiments. Any idealized experiment that modifies one (only for land) will inevitably modify the other. For this reason, we focused our experiments on the land–sea gradient, while interpreting the results in the context of the coupled role of both gradients. Furthermore, the new experiments now included in the revised manuscript clearly show that land warming is a highly relevant factor.

Regarding the Indo-Pacific warm pool, we fully agree that this region is an important contributor to tropical precipitation patterns and has strengthened in recent decades. Our analysis highlights its potential role, and the revised manuscript discusses this more explicitly in Section 3. However, designing clean perturbation experiments to isolate only the warm pool effect would require a completely different and dedicated modelling framework. This lies outside the scope of the present study, which focuses mainly on robust large-scale thermal gradients. We therefore view our discussion as a foundation and welcome future targeted experiments to further explore this question.

- V. We agree that the zero-albedo experiment does not fully separate the land–sea contrast from the inter-hemispheric temperature difference. As explained in our response to comment (iv), the two gradients are physically inseparable because the Northern Hemisphere contains more land area; any modification to land albedo only inevitably affects both gradients. Our goal was therefore not to isolate them, which is nearly an impossible task given their physical coupling, but to test whether enhancing land-surface heating—as a joint perturbation to both gradients—reproduces the observed precipitation response.
- VI. We agree that the recent increase in land temperature arises from multiple processes—including changes in surface heat fluxes, cloud radiative effects, and atmospheric longwave emissivity—and not from albedo changes alone (especially at tropical latitudes). However, our intention in the sensitivity experiments was not to identify the mechanisms that generate the land–sea thermal gradient, which are rather well-known (Toda et al., 2021, 2023), but rather to test the precipitation and circulation impacts of amplifying the gradient itself. To achieve this goal, altering land albedo offers a clean, straightforward, and minimally intrusive way to selectively enhance land heating in a physically consistent manner, without directly modifying atmospheric radiative properties or imposing artificial surface fluxes.
- VII. The land temperature and SST patterns emerge very quickly, in a few months, in both the idealized zero-albedo experiments and the new idealized desert-albedo experiments now included in the revised manuscript. In all these albedo experiments,

the Earth's Energy imbalance (e.g., the difference in the net downward shortwave radiative flux and outgoing longwave radiative flux at the top-of-atmosphere) is maximum during boreal summer and over the subtropical deserts of the Northern Hemisphere. Consistently, the dynamical response to the albedo perturbations is also the strongest during boreal summer and the land temperature and SST patterns emerge immediately after the first boreal summer in all the simulations. This demonstrates unambiguously that the land warming and the associated sensible heat flux drive the dynamical response in the Tropics and the SST changes in the simulations. In other words, land warming drives ocean changes, not the reverse.

2) The rationale behind using a simple two-layer moisture budget analysis instead of a vertically integrated moisture budget analysis was not clear. What would be the typical error for considering this simplified approach? Given that this paper is focused on precipitation, I was not expecting a simplified approach. (The use of this simplified approach in past studies cannot be the sole reason behind choosing it.)

Is there a possibility that this two-layer approach using near-surface humidity and mid-tropospheric vertical winds biases towards dynamics than thermodynamics?

Also, it seems ERA5 doesn't fully close the moisture budget. Therefore, can it impact your results and conclusions?

Answer: Our motivation for using the simplified two-layer moisture budget comes from the established physical understanding of the tropical atmosphere rather than solely on precedent works using it. Studies such as Held & Soden (2006) and Huang et al. (2013) have shown that the vertical velocity at ~500 hPa provides a robust indicator of large-scale ascent and descent throughout the troposphere. Also, 90% of the total moisture in the atmosphere is situated below 500 hPa. Thus, near-surface humidity together with mid-tropospheric vertical motion captures the leading-order thermodynamic and dynamic controls on tropical mean precipitation. As a result, this two-layer approach provides a physically meaningful first-order representation of the moisture budget where the dominant contributions occur. We acknowledge that a fully vertically integrated budget is more complete; however, our aim here is to diagnose the dominant mechanisms behind observed precipitation changes and trends, for which the simplified framework is sufficient and physically justified, as the sum of the trends in the thermodynamic and dynamic components accurately reconstructs the rainfall trends; see Fig. 2 of the revised manuscript.

Furthermore, to check the robustness of our results, we applied an additional, independent moisture-budget decomposition following Chadwick et al. (2013), adapted here for observational reanalysis (Supporting Information Fig. 5; Supporting Information Section 3). This method decomposes total precipitation change (2005–2024 minus 1981–2000) into different dynamically and thermodynamically driven components. Both the simplified moisture budget and this new decomposition consistently indicate that dynamical terms (more specifically, the shifts in the pattern of tropical convection) dominate over thermodynamic terms in explaining the observed precipitation changes. This agreement

across two independent frameworks reduces the likelihood that our conclusions are artifacts of the simplified two-layer approach.

We agree that ERA5 does not perfectly close the moisture budget. To test the robustness of our conclusions to this issue, we also repeated key elements of the analysis using alternative datasets (NCEP and MSWEP). These yield qualitatively similar results (not shown), increasing confidence that our findings do not depend on ERA5 moisture budget closure as well.

3. About climate indices (Sec 4.4): Multiple temperature-based indices were used. I was wondering if there are any reasons to not include PDO related SST change. This also brings the question whether the analysis period (1979-2024) is sufficient to distinguish between forced signals and decadal variability, given that PDO influence could be substantial.

Answer: We agree with the reviewer that the 1979–2024 period may be too short to robustly isolate PDO-related decadal variability, and for this reason, we did not include PDO-based SST indices in our analysis. This is also justified by the fact that many of the observed trends in the temperature indices are accurately reproduced in the CMIP MME of these indices, which rules out the role of the PDO for these trends. As mentioned, PDO can exert substantial influence on regional climate patterns, for example, in the Eastern Pacific, but distinguishing its internally driven decadal variability from externally forced signals requires longer observational or model-based datasets. We have also indicated this in the discussion section 3, e.g., that decadal variability, including PDO-like modes, may contribute to some of the features we observe.

4. The second half of Sec 2.4 and Section 3 (Summary and Discussion) are repetitive, and should be streamlined. But, more importantly, I felt that much of the discussion and its interpretation of the sensitivity experiments were qualitative.

Answer: We agree with this comment. To address these issues and provide a more quantitative analysis of the northward shift of the ITCZ precipitation in the idealized experiments, we first use a precipitation asymmetric index (PAI) with respect to the equator, following Hwang and Frierson (2013). The PAI is defined as precipitation in Northern Hemisphere tropics (equator to 20°N, area-averaged) minus precipitation in Southern Hemisphere tropics (equator to 20°S) normalized by the tropical mean precipitation (20°S~20°N) and assess the degree to which the ITCZ precipitation falls in the Northern Hemisphere in the idealized experiments consistent with the northward shift of the ITCZ as observed in ERA5 (Fig. 1) and also verified by the trend of the PAI in ERA5 and also noted from independent atmospheric infrared sounder observations by Aumann et al. (2024).

In addition, we also now use a Walker Circulation Index (WCI) following Vecchi et al. (2006) to illustrate the strengthening of the Pacific Walker circulation in the idealized albedo

experiments quantitatively; see our revised Section 2.4 and our answer to comment 6 below for further details.

Finally, surface and Top-Of-Atmosphere (TOA) radiation budget diagnostics are now also discussed in this revised Section 2.4 (see Fig. 6 and SI Figs. 12, 14, and 15).

All these new analyses are now discussed in Section 2.4 of the revised manuscript and confirm the northward shift of the ITCZ precipitation and the strengthening of the Pacific Walker circulation in all the idealized albedo sensitivity experiments.

5. Your regression involved rainfall and velocity potential with trends in several parameters, such as GMP, land-sea contrast, interhemispheric temperature gradient, etc. Given that many of these parameters themselves are highly correlated, it seems that one will need multivariate regression to infer the dominance of one index over another.

Answer: We acknowledge that several of the indices used in our analysis (e.g., GMP, land–sea thermal contrast, and the interhemispheric temperature gradient) are not statistically independent and therefore share a significant fraction of variance. A full multivariate regression framework would indeed be required to formally quantify the relative dominance of one index over another.

Our primary objective in the present study, however, is not to rank these indices in a strict statistical sense, but rather to demonstrate the physical consistency and robustness of the rainfall and circulation response across multiple, physically linked measures of large-scale climate change. The strong covariability among these indices reflects their common physical origin associated with anthropogenic warming and land–sea thermal contrast amplification.

Also, our approach here was that of repeating these regressions with the CMIP6 MME of the parameters to select the dominance of one index over another, and also to isolate the forced component in the relevant parameters rather than using a purely statistical multivariate approach.

6. I felt the discussion on Walker circulation strengthening/weakening was somewhat qualitative, and based on upper-level velocity potential and surface winds. Is there an index or quantitative approach to estimate the strength of Walker circulation?

Answer: To address this comment, we estimated the strength of the Walker circulation in the simulations by a Walker Circulation Index (WCI) based on sea-level pressure variations across the tropical Pacific. More precisely, this Walker circulation index is defined by the zonal contrast of sea-level pressure between the eastern (5° S– 5° N, 160° – 80° W) and western (5° S– 5° N, 80° – 160° E) equatorial Pacific and is commonly used in past studies addressing the trends of the Walker circulation in both observations and CMIP simulations, see Vecchi et al. (2006) and Watanabe et al. (2024) for details. Using this WCI index, we now illustrate quantitatively that the strength of the Walker circulation increases in both the

idealized zero-albedo and desert-albedo experiments compared to the control runs. As an illustration, the strength of the Walker circulation in the zero-albedo experiments increases by 217% and 117% relative to MODIS-ctrl runs and by 192% and 112% relative to ctrl runs, in the CFS and SINTEX models, respectively.

7. *Overall presentations need significant improvements throughout the paper, such as:*

(i) *Subsections in “Results” are named as “Trend Analysis” (sec 2.1), “Time Series and Regression Analysis” (sec 2.3) – I would avoid naming sections following statistical methods. Instead, sections should be named after the scientific content or processes discussed in that section.*

(ii) *It was very difficult to go through several figures in the manuscript as well as in the supporting document because of various reasons, like very small fonts, small panel sizes, etc. Figures need to be thoroughly modified for improving clarity.*

(iii) *Please add units to the figure captions. Check all.*

Answer: We thank the reviewer for these helpful suggestions regarding the overall presentation of the manuscript. We have carefully revised the manuscript to improve clarity, organization, and figure readability, as detailed below.

(i) We agree with the reviewer that subsection titles should emphasize scientific content rather than statistical methods. Accordingly, “Trend Analysis” has been renamed to “Global Climate Trends”, and “Time Series and Regression Analysis” has been renamed to “Possible Drivers of Precipitation and Atmospheric Circulation Trends.” These revised titles better reflect the physical processes and scientific interpretation discussed in each section.

(ii) We have thoroughly revised the figures in both the main manuscript and the Supplementary Information to improve readability and clarity. These revisions include increasing font sizes and improving overall layout and labelling. We believe that these changes enhance the clarity of the figures. If any specific figures remain unclear, we would be grateful if the reviewer could indicate the particular figure(s) and issue(s), and we would be happy to make further improvements.

(iii) Units have now been added to all figure captions, and all captions have been carefully checked for consistency and completeness.

8. *There were discussions about dynamic and thermodynamic controls on changes in precipitation – this is prevalent in the literature as well. I was curious to know to what extent dynamic and thermodynamic effects are independent of each other!! Is it possible to separate their influences in sensitivity experiments?*

Answer: We agree that this question makes sense for the idealized zero-albedo experiments discussed in our original submission, as the large land surface warming induced by the strong

albedo perturbation in these runs can potentially induce a large moistening of the lower atmosphere over both land and ocean.

However, the strong consistency of these original idealized zero-albedo experiments with the new idealized desert-albedo experiments (now discussed in the revised manuscript in Section 2.4) in which a much more modest albedo perturbation is now imposed only over the subtropical deserts of the Northern Hemisphere clearly demonstrate that the simulated precipitation changes are almost exclusively driven by the dynamics as the near surface relative humidity over these arid regions is essentially zero.

9. Apart from the surface forcing (land-sea temperature contrast), I would imagine that the atmospheric stability over land and ocean would also play a role, but I did not seem to find anything on stability – any thoughts?

Answer: As for question (9) above, the new idealized desert-albedo experiments discussed in the revised manuscript shed some light on the possible role of atmospheric stability, especially for the response of the African and Asian summer monsoons in the idealized experiments. In short, both of these monsoon systems are adjacent to the NH subtropical deserts over which the albedo perturbations are imposed in the runs. Interestingly, the associated warming of the lower atmosphere over the deserts is able to destabilize the atmosphere at the margins of these monsoon systems and to trigger a large rainfall increase for both the West African and Indian monsoons in the runs, see SI Figs. 13-15 and Sooraj et al. (2019) for more details on this question.

10. Observed precipitation trend (1979-2024) in the tropics (Fig. 1a) seems to be statistically insignificant. What does it mean in the context of the sensitivity experiments that show the simulated precipitation trend is largely statistically significant (Fig. 5)?

Answer: First, we note that the observed precipitation trend (1979-2024) in the tropics, as illustrated in Fig. 1a are not statistically insignificant elsewhere as suggested by the reviewer. The rainfall increases over India and the Maritime Continent, the drying over east Asia, and the northward shift and narrowing of the ITCZ, especially in the Pacific, are all significant in Fig. 1a. We hope that the improved Fig. 1a will clearly illustrate that now. Furthermore, these features are also significant in the idealized zero-albedo experiments (see Fig. 5b of the revised manuscript), which adds consistency to our interpretation. Next, the rainfall responses in the new idealized desert-albedo experiments (SI Fig 13) are still statistically significant despite the amplitude of the response being substantially decreased compared to the zero-albedo experiments (as expected) and are now more in line with the observations. Furthermore, the rainfall patterns are again similar to those found in observations and the idealized zero-albedo experiments. This again demonstrates the robustness of our results and that our original interpretation of the results is sound.

Other comments:

Introduction: 1st line: State of Climate to State of the Climate

2nd line: mean surface air temperature to mean near-surface air temperature

3rd paragraph, first sentence: “Taken together” The sentence is not clear, rewording needed.

Paragraph starting with “In this context...” and the next paragraph: very long, consider splitting. Check all.

Page 3, line 6: theories to hypotheses

Page 3, 2nd para, line 11: remove “actually”

Page 4, line 3: sensitivity coupled experiments to coupled sensitivity experiments

Answer: All done! Thank you for pointing this out.

Page 4, sec 2.1, line 8: SST trends or SST patterns

Answer: We are sticking with SST trends since it is the SST trends that are showing a La-Nina-like pattern.

Page 5: (d) 850-hPa wind trend (e) 200-hPa velocity potential trend

Wind vectors are not legible at all.

Answer: We do not think it is necessary to add ‘trend’ in (d) and (e) since the first line mentions ‘Global climate trends’ in (a) and so on. The wind vectors have now been updated as requested.

Page 6, line 7: It seems Supp Fig. 4 is introduced before Supp fig. 3. Check.

Page 6, Line 13: “Yet, importantly,” Not clear, rewording needed.

Page 6, line 11 from bottom: also arises to may arise

Page 6, line 14 from bottom: The sentence, “As this contrast” is very long. Consider splitting.

Answer: All done! Thank you for pointing this out.

Page 6, line 11 from below: what is “effective radiative forcing” here?

Answer: In this context, “effective radiative forcing” or ERF has its standard meaning: the net radiative perturbation induced by CO₂ after fast atmospheric adjustments have taken place but before the surface temperature responds. Toda et al. (2021) and other studies on the land–sea thermal contrast have shown that spatial differences in ERF play a significant role in driving the stronger warming over land.

Page 7, lines 2-3: However, the Southern Oceanremove because you are talking about it just a few sentences later.

Page 8, line 5: In summary to Overall

Answer: All done! Thank you for these comments.

Page 8, line 12 from bottom: “However, the reconstructed trend” – Was not clear which panels are being compared

Page 11, 2nd para, “Its regression” – not clear which regression

Answer: The sentences have been restructured for clarity. Thank you for noticing this.

Page 13, line 8 from bottom: “existence of feedbacks...” – I was curious what these feedbacks may be.

Answer: The feedbacks likely include the same physical mechanisms that contribute to the land–sea thermal contrast, such as differences in lapse-rate, water-vapour, and cloud responses, and these effects are amplified by the fact that the Northern Hemisphere contains substantially more land than the Southern Hemisphere.

Page 16, sec 3, 2nd para: “ These results suggest three possible scenarios:” Can they all be true?

Answer: Yes, that is possible. But it would require more modelling studies to further investigate this.

Page 21, “by fitting a polynomial of degree 1 or 2” – any reason not to consider polynomials of higher degrees? Also, I see a window width of 6 years. Does it remove/smooth ENSO or decadal variability?

Answer: The smoothness of the trend time series as estimated by the LOESS method, summarized in Section 4.6 of the revised manuscript, is governed by two factors: the size of the (time) moving window used for local fitting and the degree of the polynomial that is fitted in this time window. The trend is smoother when the width of the window increases or when the degree of the polynomial decreases. As soon as we use polynomials of higher degrees, the smoothness of the trend decreases, and at the limit, when the degree of the polynomial is equal to the width of the window, the resulting trend time series is equal to the original time series! In other words, fitting a polynomial of degree 1 or 2 ensures a certain degree of smoothness of the trends, which is exactly what we seek here.

Here, effectively we use a window width of 6 years (used 4 years as well and the results remain unchanged (not shown)) as stated in Section 4.5; this is sufficient to remove decadal variability in all the cases, but the observed Niño 3-4 index and EEPO SST time series as seen in Fig. 3d and g, which both show an oscillatory behaviour superposed on an increasing warming trend. Remarkably, the observed IPWP SST time series does not show such oscillatory behaviour and follows exactly the same trend pattern as the CMIP IPWP SST time series, despite the same LOESS parameters being used in all the fittings.

Supplementary Fig. 4: Did you use the ONI index to separate EL Niño and La Niña phases? Oftentimes, the phases change within the same year, so considering a full year as an El Niño or La Niña year may not be well-suited here.

Also, mention how the inter-hemispheric thermal gradient was calculated (NH minus SH or SH minus NH).

Answer: Yes, we used the ONI index to identify El Niño and La Niña years. We acknowledge that ENSO phases can change within a year; however, many previous studies have used a variety of approaches to construct ENSO composites, such as focusing only on peak ENSO months, using seasonal classifications, or defining ENSO years differently using a different index, and these methods all produce broadly consistent composite patterns. For example, Kim et al. (2016, their Fig. 4) show July(0)–June(1) ENSO composites based on peak-phase conditions, and the resulting spatial patterns closely resemble those in our analysis. Since we use ENSO years solely for constructing composites and not for diagnosing sub-annual variability, and given that our composites are consistent with those from alternative methods in the literature, we believe our approach is sufficiently robust for Supplementary Fig. 3 of the revised manuscript (e.g., old SI Fig. 4).

We also clarify in the manuscript that the inter-hemispheric thermal gradient is calculated as NH minus SH.

References that are not in the main text but cited here:

Kim, J., J. Kug, J. Yoon, and S. Jeong, 2016: Increased Atmospheric CO₂ Growth Rate during El Niño Driven by Reduced Terrestrial Productivity in the CMIP5 ESMs. *J. Climate*, **29**, 8783–8805, <https://doi.org/10.1175/JCLI-D-14-00672.1>.

Reply to Reviewer 3:

*The authors study a range of climate responses using ERA5 reanalysis, CMIP6 simulations, targeted sensitivity experiments by GCMs with a focus on tropical precipitation responses. They motivate their study by current climate models which show large precipitation biases in the tropics. They also describe several current “theories”, among these “wet-get-wetter” (WeGW) and “warm-get-wetter” paradigms. Within WeGW and weakening of atmospheric circulations, e.g., Walker and Hadley cells, due to increased atmospheric moisture and reduced radiative cooling is postulated. They point out that there are discrepancies between model and observational findings, where the latter show a strengthening of the Walker circulation despite significant surface warming. Also, CMIP models show a decrease in the equatorial SST gradient during recent decades, whereas observations show an increase. Citing Shrestha, S., Soden, B.J.: Anthropogenic weakening of the atmospheric circulation during the satellite era. *Geophysical Research Letters* 50(22) (2023) (Ref 39) the authors state that global mean circulation might indeed be weakening due to reduction in the Walker circulation. Joseph et al. further caution that precipitation response to global warming is a complex phenomenon, with an ongoing debate on whether circulation changes might be a main cause. They acknowledge that, how the water cycle might have changed in the course of recent decades, remains poorly known. In their results they distinguish three main sections:*

- 1. A discussion on spatial trends in key variables: precipitation, 2m air temperature, SST, 850 hPa winds, as well as 200hPa velocity;*
- 2. They then decompose precipitation changes into dynamic and thermodynamic components;*
- 3. They finally discuss eight surface temperature indices in observations and simulations. In detail, within their trend analysis they find that ERA reanalysis data roughly agrees with other observational and reanalysis datasets regarding overall trends. CMIP6 model simulated precipitation shows very different trends from the ERA data, e.g., with weak temperature changes in the tropics but strong precipitation changes.*

Pasted figure 1: ERA-derived precipitation changes

Pasted figure 2: CFS (left column) and SINTEX (right column)

Regarding the attribution to dynamics vs. thermodynamics, in the following subsection they demonstrate that the observed tropical rainfall trends mainly appear to be controlled by dynamical processes rather than thermodynamics. They claim that surface temperature gradients play a key role in driving the atmospheric circulation. They finally discuss several indices, even though the necessity for this discussion did not become obvious to me, it seems not needed here. More interestingly, they attempt to attribute the precipitation changes seen in ERA data to other variables, in particular near-surface temperature. They isolate trends in land-sea warming contrast as a potential cause of circulation changes and thus precipitation pattern shifts. They further use two global coupled models, namely CFS and SINTEX to re-enact (and likely exaggerate) the effect of land-sea temperature changes by reducing the over-land albedo to zero. Indeed, in some regions the patterns of precipitation changes

produced by their modelling results are not too different from the ERA-derived results (see the comparison of the panels in Fig. 1a vs. Materials Fig. 2c,d). This is especially true for precipitation over the Pacific Ocean where some notable similarities exist. Yet, in most other regions of the globe the changes in precipitation seem much less similar and are often even of opposite sign, e.g. most continental regions and the Atlantic Ocean.

In summary, I agree that CMIP models do not show a strong similarity with ERA-derived historical precipitation trends. I appreciate that the authors attempted an idealized sensitivity study using their own model simulations to emphasize certain aspects of precipitation change patterns. I am currently not convinced that the changes they simulate sufficiently describe the ERA-derived changes in precipitation, even though some similarity exists for the Pacific Ocean -but most other regions disagree.

Answer: We thank the reviewer for the detailed summary of our study and for the comments. We understand the concern that the precipitation changes simulated in our sensitivity experiments do not fully match the ERA5-derived trends.

We would like to clarify that our intention was not to claim a one-to-one reproduction of the observed patterns, nor do we argue that the land–sea (or hemispheric) thermal gradient explains all aspects of the observed precipitation changes. Our goal was to test whether altering the land–sea thermal contrast can generate some of the key large-scale precipitation features seen in observations.

Our original and new sensitivity experiments (e.g., see Table 1 of the revised manuscript) do reproduce several broad patterns, particularly the wetting north of the equator, wetting over the western Pacific, wetter conditions over the northern Indian region, and the drying south of the equatorial Pacific. These features are consistent with the dynamical adjustments we highlight. We also explicitly acknowledge in the revised manuscript, both in Sections 2.4 and 3 (p. 15, second paragraph), that notable regional discrepancies remain in the Atlantic sector, and that other processes and modes of variability are likely contributing to the full pattern, like the strengthening of the Indo-Pacific warm pool (Weller et al., 2016). For example, in Africa, even the observational datasets themselves show disagreement (ERA5 vs. GPCP; see Fig. 1a and Supplementary Fig. 1a), making it unsurprising that idealized model experiments do not capture the full complexity of regional precipitation changes in such areas as well.

We additionally clarify that the reviewer’s referenced figure comparison appears to be a misunderstanding. The comment refers to “Fig. 1a vs. Materials Fig. 2c,d,” which corresponds to ERA precipitation trends versus CMIP6 simulations. The appropriate comparison for our sensitivity experiments is Fig. 1a versus Fig. 5c,d in the old manuscript, where the effects of the land-albedo perturbations are shown. As discussed in the manuscript, these experiments reproduce several large-scale features but are not expected to match all regional details.

To strengthen our arguments, we now also discuss new experiments (the desert-albedo experiments; see Table 1 and Sections 2.4 and 4.7 of the revised manuscript) as suggested by the other reviewers, which are consistent with our original results and also include relevant surface and Top-Of-Atmosphere (TOA) radiation budget diagnostics in the revised manuscript. We hope that this will be more convincing for the reviewer.

Reply to Reviewer:

2nd round review of “Tropical precipitation response to anthropogenic climate change in recent decade” by Joseph et al. for publication in Nature Communications.

The authors have adequately addressed my main concerns. Here are a few suggestions that could further improve the clarity of the paper.

Answer: We thank the reviewer for his/her positive assessment and for confirming that the main concerns have been adequately addressed. We also appreciate the additional suggestions, which have helped further improve the clarity of the manuscript.

1. (ii) I believe that further experiments have provided more confidence on the results. In this regard, the authors argue, “We believe that this strategy is the right one in the present context as we deal only with mean-state change, not variability.” – This statement seems to hold only if variability doesn’t influence the evolving mean-state. However, in a changing climate, the mean state and variability may evolve together, and variability could potentially feed back onto mean state changes. So, the question is, how do you know that changes in variability do not affect the evolving mean state in this context?

Answer: Thank you for acknowledging that further experiments have provided more confidence in the results. We agree that mean state and variability may evolve together in a changing climate. In this study, we focus on the forced mean-state response to strong and idealized perturbations that produce rapid and stable adjustments, and we assess robustness through the close consistency of responses across two structurally different coupled models and multiple sensitivity experiments. However, interactions between variability and the evolving mean state indeed remain an important topic but lie beyond the scope of the present analysis.

1. (iv) Part of your response to comment (iv) would be helpful to include in the paper for improved clarity.

Answer: Thank you for this suggestion. We have now incorporated these points directly into the Summary and Discussion section (page 21, paragraph 2) to improve clarity and interpretation.

1. (v) Same here. A sentence or two related to why isolation was not the goal should be included in the paper.

Answer: Thank you again for this suggestion. We have now included these points in the summary and discussion section (page 21, paragraph 2).

1. (vii) The authors state: “The land temperature and SST patterns emerge very quickly, in a few months, in both the idealized zero-albedo experiments and the new idealized desert-albedo experiments now included in the revised manuscript.” – What I was looking for was if the land temperature patterns emerged earlier than SST patterns.

If land and SST patterns emerge simultaneously, how can land surface warming be interpreted as the driver of SST changes?

Answer: Thank you for this important clarification. In our experiments, land temperature changes emerge rapidly, within days to weeks, whereas the SST response develops later. This temporal sequencing supports the interpretation that land surface warming acts as the driver of the subsequent SST changes. We have now made this distinction explicit in the manuscript (page 20, last paragraph; page 15, paragraph 3).